# ControlMM: Controllable Masked Motion Generation

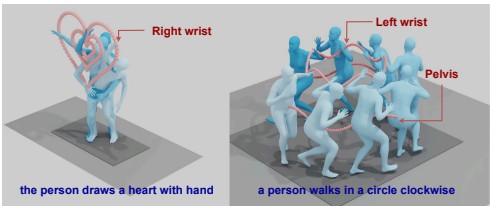 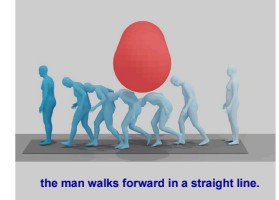 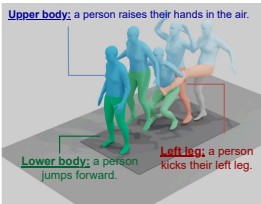

**(a) Any Joint Any Frame Control**   **(b) Obstacle Avoidance for All Joints**   **(c) Body Part Timeline Control**

Figure 1: ControlMM enables a wide range of applications in text-to-motion generation with high quality and precision. (a) Any Joint, Any Frame Control: spatial control signals for specific joints and frames. (b) Object Avoidance for All Joints: generates motion that avoids obstacles for any joint. (c) Body Part Timeline Control: generates motion from multiple text prompts, each corresponding to different body parts.

## Abstract

Recent advances in motion diffusion models have enabled spatially controllable text-to-motion generation. However, despite achieving acceptable control precision, these models suffer from generation speed and fidelity limitations. To address these challenges, we propose ControlMM, a novel approach incorporating spatial control signals into the generative masked motion model. ControlMM achieves real-time, high-fidelity, and high-precision controllable motion generation simultaneously. Our approach introduces two key innovations. First, we propose masked consistency modeling, which ensures high-fidelity motion generation via random masking and reconstruction, while minimizing the inconsistency between the input control signals and the extracted control signals from the generated motion. To further enhance control precision, we introduce inference-time logit editing, which manipulates the predicted conditional motion distribution so that the generated motion, sampled from the adjusted distribution, closely adheres to the input control signals. During inference, ControlMM enables parallel and iterative decoding of multiple motion tokens, allowing for high-speed motion generation. Extensive experiments show that, compared to the state of the art, ControlMM delivers superior results in motion quality, with better FID scores (0.061 vs 0.271), and higher control precision (average error 0.0091 vs 0.0108). ControlMM generates motions 20 times faster than diffusion-based methods. Additionally, ControlMM unlocks diverse applications such as any joint any frame control, body part timeline control, and obstacle avoidance. Video visualization can be found at `https://anonymous-ai-agent.github.io/CAM`

## 1 Introduction

Text-driven human motion generation has recently gained significant attention due to the semantic richness and intuitive nature of natural language descriptions. This approach has broad applications in animation, film, virtual/augmented reality (VR/AR), and robotics. While text descriptions offer a wealth of semantic guidance for motion generation, they often fall short in providing precise spatial control over specific human joints, such as the pelvis and hands. As a result, achieving natural interaction with the environment and fluid navigation through 3D space remains a challenge.

To tackle this challenge, a few controllable motion generation models have been developed recently to synthesize realistic human movements that align with both text prompts and spatial control signals Shafir et al. (2023); Rempe et al. (2023); Xie et al. (2023); Wan et al. (2023). However, existing solutions face significant difficulties in generating high-fidelity motion with precise and flexible spatial control while ensuring real-time inference. In particular, current models struggle to support both sparse and dense spatial control signals simultaneously. For instance, some models excel at generating natural human movements that traverse sparse waypoints Karunratanakul et al. (2023); Rempe et al. (2023), while others are more effective at synthesizing motions that follow detailed trajectories specifying human positions at each time point Wan et al. (2023). Recent attempts to support both sparse and dense spatial inputs encounter issues with control precision; the generated motion often is not aligned well with the control conditions Xie et al. (2023). Besides unsatisfied spatial flexibility and accuracy, the quality of motion generation in controllable models remains suboptimal, as evidenced by much worse FID scores compared to models that rely solely on text inputs. Moreover, most current methods utilize motion-space diffusion models, applying diffusion processes directly to raw motion sequences. While this design facilitates the incorporation of spatial control signals, the redundancy in raw data introduces computational overhead, resulting in slower motion generation speeds.

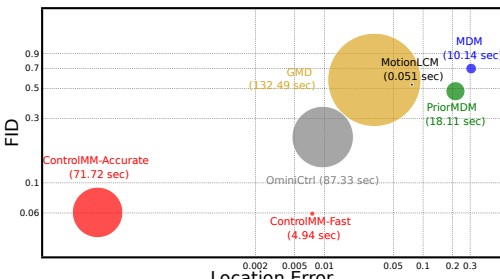

Figure 2: Comparison of FID score, spatial control error, and motion generation speed (circle size) for our accurate and fast models comparing to state-of-the-art models. The closer the point is to the origin and the smaller the circle, the better performance.

To address these challenges, we present ControlMM, a novel approach that integrates spatial control signals into generative masked motion models that excels in high-quality and fast motion generation Pinyoanuntapong et al. (2024b); Guo et al. (2023); Pinyoanuntapong et al. (2024a). ControlMM is the first method capable of achieving real-time, high-fidelity, and high-precision controllable motion generation simultaneously. Our contributions can be summarized as follows. (1) We introduce masked consistency modeling, the first approach that incorporates spatial guidance into Masked Motion Model, which results in higher generation quality, more precise control, accelerated generation, and broader applications compared to existing methods as shown in Fig. 1. (2) We propose an inference-time logit-editing approach, which strikes the optimal balance between inference time and control precision, while enabling new control tasks, such as obstacle avoidance, in a zero-shot manner. (3) We conduct extensive qualitative and quantitative evaluations on multiple tasks. As shown in Fig. 2, our model outperforms current state-of-the-art methods in motion generation quality, control precision, and speed with multiple applications *i.e.* joint-specific control, obstacle avoidance, body part timeline control.

## 2 RELATED WORK

**Text-driven Motion Generation.** Early methods for text-to-motion generation primarily focus on aligning the latent distributions of motion and language, typically by employing loss functions such as Kullback-Leibler (KL) divergence and contrastive losses. Representative works in this domain include Language2Pose (Ahuja & Morency, 2019), TEMOS (Petrovich et al., 2022), T2M (Guo et al., 2022b), MotionCLIP (Tevet et al., 2022a), and DropTriple (Yan et al., 2023). However, the inherent discrepancy between the distribution of text and motion often results in suboptimal generation quality when using these latent space alignment techniques.

Recently, diffusion models have become a widespread choice for text-to-motion generation, operating directly in the motion space (Tevet et al., 2022b; Zhang et al., 2022; Kim et al., 2022), VAE latent space (Chen et al., 2022), or quantized space (Lou et al., 2023; Kong et al., 2023). In these works, the model gradually denoises the whole motion sequence to generate the output in the reverse diffusion process. Another line of work explores the token-based models in the human motion domain, for example, autoregressive GPTs (Guo et al., 2022a; Zhang et al., 2023a; Jiang et al., 2023; Zhong et al., 2023) and *masked motion modeling* (Pinyoanuntapong et al., 2024b;a; Guo et al., 2023). These

methods learn to generate discrete motion token sequences that are obtained from a pretrained motion VQVAE (Esser et al., 2020; Williams et al., 2020). While GPT models usually predict the next token from history tokens, masked motion models utilize the bidirectional context to decode the masked motion tokens. By predicting multiple tokens at once, the masked modeling methods can generate motion sequences in as few as 15 steps, achieving state-of-the-art performance on generation quality and efficiency. Despite the performance gains of masked motion models, supporting spatial controllability in these models remains unexploited. This paper is the first work that proposes controllable masked motion model to simultaneously achieve high-quality motion generation with high-precision spatial control.

**Controllable Motion Synthesis.** In addition to text prompts, synthesizing motion based on other control signals has also been a topic of interest. Example control modalities include music (Li et al., 2021b;a; Lee et al., 2019; Siyao et al., 2022; 2023; Tseng et al., 2022), interacting object (Kulkarni et al., 2024; Diller & Dai, 2024; Li et al., 2023; Cha et al., 2024), tracking sensors (Du et al., 2023), scene (Huang et al., 2023; Wang et al., 2024) programmable motion (Liu et al., 2024), style (Zhong et al., 2024), goal-reaching task (Diomataris et al., 2024), and multi-Track timeline Control (Petrovich et al., 2024). Peng et al. (2021; 2022); Xie et al. (2021); Yuan et al. (2022); Luo et al. (2023a;b); Tessler et al. (2024) incorporate physics to motion generation. To control the trajectory, PriorMDM (Shafir et al., 2023) finetunes MDM to enable control over the locations of end effectors. CondMDI (Cohan et al., 2024) generates motion in-betweening from arbitrarily placed dense or sparse keyframes. GMD (Karunratanakul et al., 2023) and Trace and Pace (Rempe et al., 2023) incorporates spatial control into the diffusion process by guiding the root joint location. OmniControl (Xie et al., 2023) extends the control framework to any joint, while MotionLCM (Dai et al., 2024) applies this control in the latent space, both leveraging ControlNet (Zhang et al., 2023b). DNO (Karunratanakul et al., 2024) introduces an optimization process on the diffusion noise to generate motion that minimizes a differentiable objective function. Recent approaches (Wan et al., 2023; Huang et al., 2024) model each body part separately to achieve fine-grained control but are limited to dense trajectory objectives.

## 3 CONTROLMM

The objective of ControlMM is to enable controllable text-to-motion generation based on a masked motion model that simultaneously delivers high precision, high speed, and high fidelity. In particular, given a text prompt and an additional spatial control signal, our goal is to generate a physically plausible human motion sequence that closely aligns with the textual descriptions, while following the spatial control conditions, i.e., $(x, y, z)$ positions of each human joint at each frame in the motion sequence. Towards this goal, in Section 3, we first introduce the background of conditional motion synthesis based on the generative masked motion model. We then describe two key components of ControlMM, including masked consistency training in Section 3.2 and inference-time logits editing in Section 3.3. The first component aims to learn the categorical distribution of motion tokens, conditioned on spatial control during training time. The second component aims to improve control precision by optimally modifying learned motion distribution via logits editing during inference time.

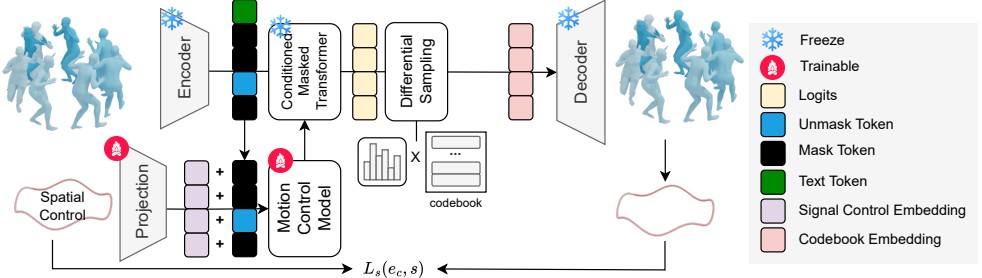

Figure 3: Training phase of ControlMM, the pretrained *Encoder*, *Decoder* and *Conditioned Masked Transformer* are frozen, only the *Motion Control Model* is trained.

### 3.1 PRELIMINARY: GENERATIVE MASKED MOTION MODEL

*Masked Motion Models* generally consist of two stages : Motion Tokenizer and Text-conditioned Masked Transformer  (Pinyoanuntapong et al., 2024b;a; Guo et al., 2023).  The objective of the Motion Tokenizer is to learn a discrete representation of motion by quantizing the encoder's output embedding $z$ into a codebook $\mathcal{C}$. For a given motion sequence $\mathcal{P} = [p_1, p_2, ..., p_F]$, where each frame $p$ represents a 3D pose, Motion Tokenizer outputs a discrete motion tokens $X = [x_1, x_2, ..., x_L]$. Specifically, the encoder compresses $\mathcal{P}$ into a latent embedding $z \in \mathbb{R}^{t \times d}$ with a downsampling rate of $F/L$. The embedding $z$ is quantized into codes $c \in \mathcal{C}$ from the codebook $\mathcal{C} = \{c_k\}_{k=1}^{K}$, which contains $K$ codes. The nearest code is selected by minimizing the Euclidean distance between $z$ and the codebook entries, computed as $\hat{z}_i = \operatorname{argmin}_j \|\mathbf{z} - c_j\|_2^2$. The vector quantization loss $L_{VQ}$ is defined as:

$$L_{VQ} = \| \operatorname{sg}(\mathbf{z}) - \mathbf{c}\|_2^2 + \beta \|\mathbf{z} - \operatorname{sg}(\mathbf{c})\|_2^2, \tag{1}$$

where $\operatorname{sg}(\cdot)$ is the stop-gradient operator and $\beta$ is a hyper-parameter for commitment loss.

During the second stage, the quantized motion token sequence $X = [x_1, x_2, ..., x_L]$ is updated with [MASK] tokens to form the corrupted motion sequence $X_{\overline{\mathbf{M}}}$. This corrupted sequence along with text embedding $W$ are fed into a text-conditioned masked transformer parameterized by $\theta$ to reconstruct input motion token sequence with reconstruction probability equal to $p_\theta \left( x_i \mid X_{\overline{\mathbf{M}}}, W \right)$, which is obtained by the motion token classifier.  The objective is to minimize the negative log-likelihood of the predicted masked tokens conditioned on text:

$$\mathcal{L}_{\text{mask}} = - \mathbb{E}_{\mathbf{X} \in \mathcal{D}} \left[ \sum_{\forall i \in [1, L]} \log p \left( x_i \mid X_{\overline{\mathbf{M}}}, W \right) \right]. \tag{2}$$

During inference, the transformer masks out the tokens with the least confidence and predicts them in parallel in the subsequent iteration. The number of masked tokens $n_M$ is controlled by a masking schedule, a decaying function of the step $t$. Early iterations use a large masking ratio due to high uncertainty, and as the process continues, the ratio decreases as more context is available from previous predictions.

### 3.2 MOTION CONTROL MODEL

ControlMM aims to generate a human motion sequence based on the text prompt (W) and spatial control signal (S). Towards this goal, we introduce a masked consistency modelling approach, which aims to learn the motion token distribution jointly conditioned on $W$ and $S$ by exploiting conditional token masking with consistency feedback.

**Conditioned Masked Transformer with Motion Control Model.** We design a masked transformer architecture to learn the conditional motion token distribution. This is the first attempt to incorporate the ControlNet design principle (Zhang et al., 2023b) from diffusion models into generative masked models, such as BERT-like models for image, video, language, and motion generation (Devlin et al., 2019; Chang et al., 2022; 2023; Villegas et al., 2022). Our architecture consists of a pre-trained text-conditioned masked motion model and a motion control model. The pre-trained model provides a strong motion prior based on text prompts, while the motion control model introduces additional spatial control signals. Specifically, the motion control model is a trainable replica of the pre-trained masked motion model, as shown in Fig 3. Each Transformer layer in the original model is paired with a corresponding layer in the trainable copy, connected via a zero-initialized linear layer. This initialization ensures that the layers have no effect at the start of training. Unlike the original masked motion model, the motion control model incorporates two conditions: the text prompt $W$ from the pre-trained CLIP model (Radford et al., 2021) and the spatial control signal $S$. The text prompt $W$ influences the motion tokens through attention, while the spatial signal $S$ is directly added to the motion token sequence via a projection layer.

**Generative Masking Training with Consistency Feedback.** The conditioned masked transformer is trained to learn the conditional distribution $p_\theta \left( x_i \mid X_{\overline{\mathbf{M}}}, W, S \right)$ by reconstructing the masked motion tokens, conditioned on the unmasked tokens $X_{\overline{\mathbf{M}}}$, text prompt (W), and spatial control signal (S). The spatial control condition is a sequence of joint control signals $S = [s_1, s_2, ..., s_F]$ with

$s_i \in \mathbb{R}^{j \times 3}$. Each control signal $s_i$ specifies the targeted 3D coordinates of the joints to be controlled, among the total $j$ joints, while joints that are not controlled are zeroed out. Since the semantics of the generated motion are primarily influenced by the textual description, to guarantee the controllability of spatial signals, we extract the spatial control signals from the generated motion sequence and directly optimize the consistency loss between input control signals and those extracted from the output. This consistency training not only enhances controllability but also addresses a unique challenge in controllable motion generation. In the image domain, spatial control signals can be directly applied, and uncontrolled regions are simply zeroed out. However, for motion control, zero-valued 3D joint coordinates are ambiguous: they may indicate that a joint is controlled with its target position at the origin in Euclidean space, or that the joint is uncontrolled. To resolve this ambiguity, we concatenate the spatial control signal with the relative difference between the control signal and the generated motion, forming the final spatial control guidance $s$. Please refer to Section A.9 for more details.

**Training-time Differential Sampling.** While consistency training offers significant benefits, integrating consistency loss into the training of generative masked models presents a challenge: the need to convert discrete motion tokens in the latent space into motion representations in Euclidean space. This conversion requires sampling from the categorical distribution of motion tokens during training, a process that is inherently non-differentiable. To address this, we leverage the straight-through Gumbel-Softmax technique (Jang et al., 2017). This approach performs categorical sampling during the forward pass and approximates the categorical distribution with differentiable sampling using the continuous Gumbel-Softmax distribution during the backward pass, i.e.,

$$p_\theta \left( x_i \mid X_{\overline{\mathbf{M}}}, W, S \right) = \frac{\exp \left( (\ell_i + g_i)/\tau \right)}{\sum_{j=1}^k \exp \left( \ell_j/\tau \right)}, \tag{3}$$

where $l$ is logits, $\tau$ refers to temperature, and $g$ represents Gumbel noise with $g_1, \ldots, g_k$ being independent and identically distributed (i.i.d.) samples from a Gumbel$(0, 1)$ distribution. The Gumbel$(0, 1)$ distribution can be sampled via inverse transform sampling by first drawing $u \sim \text{Uniform}(0, 1)$ and then computing $g = -\log(-\log(u))$.

**Motion Consistency Loss.** With the help of the training-time differential sampling, we are able to define the consistency loss, which assesses how closely the joint control signal extracted motion the generated motion aligns with the input spatial control signal $s$:

$$L_s(e_c, s) = \frac{\sum_n \sum_j \sigma_{nj} \odot \| s_{nj} - R(D(e_c)) \|}{\sum_n \sum_j \sigma_{nj}}, \tag{4}$$

where $\sigma_{nj}$ is a binary value indicating whether the spatial control signal $s$ contains a control value at frame $n$ for joint $j$. The motion tokenizer decoder $D(\cdot)$ converts motion embedding into relative position in local coordinate system and $R(\cdot)$ further transforms the joint's local positions to global absolute locations. The global location of the pelvis at a specific frame can be calculated from the cumulative aggregation of rotations and translations from all previous frames. The locations of the other joints can also be computed by the aggregation of the relative positions of the other joints to the pelvis position. The final loss for masked consistency training is the weighted combination masked training loss and motion consistency loss:

$$\mathcal{L} = \alpha \mathcal{L}_{\text{mask}} + (1 - \alpha) L_s(e_c, s). \tag{5}$$

## 3.3 INFERENCE-TIME LOGITS AND CODEBOOK EDITING

The goal of inference-time editing is to enhance control precision by further reducing the discrepancy between the generated motion and the desired control objectives. This approach does not require pretraining on specific spatial control signals, allowing the model to handle arbitrary, out-of-distribution spatial signals during inference, enabling new control tasks such as obstacle avoidance in a zero-shot manner.

The core idea behind logits editing is to update the learned logits through gradient-guided optimization during inference, allowing manipulation of the conditional motion distribution. This ensures that the generated motion, sampled from the adjusted distribution, aligns closely with the input control signals. The optimization process is initialized with the logits obtained from masked consistency

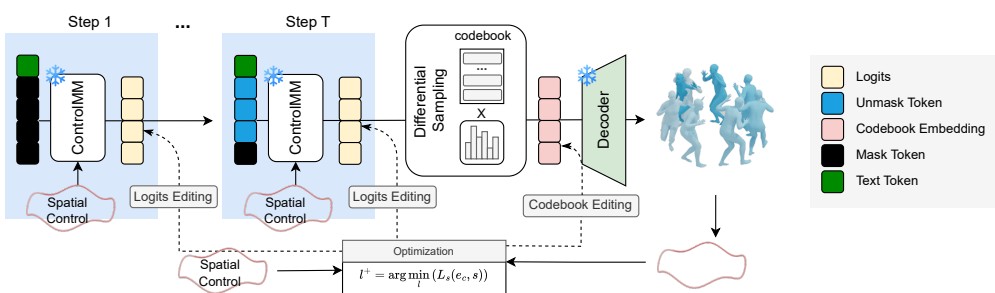

Figure 4: Inference of ControlMM. Spatial control is added to the model as input. The output logits are reconstructed and optimized through *Differentiable Sampling* in each iteration.

training, and these logits are iteratively updated to minimize the consistency loss.

$$l^+ = \arg\min_l \left(L_s(e_c, s)\right). \tag{6}$$

At each iteration $i$, the logits $l_i$ are updated using the following gradient-based approach:

$$l_{i+1} = l_i - \eta \nabla_{l_i} L_s(l_i, s). \tag{7}$$

where $\eta$ controls the magnitude of the updates to the logits, while $L_s(l_i, s)$ represents the gradient of the objective function with respect to the logits $l$ at iteration $i$. This refinement process continues over $I$ iterations. Similarly, in the last unmask step, optimizing embeddings from the codebook space is possible since there is no need to pass them to the Masked Transformer. **Codebook Editing** can further optimize the embedding in motion codebook to minimize the consistency loss:

$$e_c^{i+1} = e_c^i - \eta \nabla_{e_c^i} L_s(e_c^i, s), \tag{8}$$

where $e_c$ represents the embedding in the codebook space. Our experiments demonstrate that combining joint logits and codebook editing results in the best performance. More details about the challenges of guidance in Masked Transformers can be found in Section A.10.

## 4 APPLICATIONS

**Any Joints Any Frame Control .** To control specific joints at particular frames, the spatial control signal can be directly applied to the desired joint and frame in the global position, as the loss function during training is specifically designed for this task.

**Obstacle Avoidance.** Since *inference-time logits and codebook editing* is versatile, it can be compatible with arbitrary loss function. The Signed Distance Function (SDF) can serve as a loss function for obstacle avoidance, where the gradient field dictates the direction to repel from obstacles. This loss function incorporates a safe distance threshold $d$, beyond which the gradient diminishes to zero, and is defined as:

$$\mathcal{L}_{\text{obs}}(x) := \sum_{i,n} -\min\left[\text{SDF}\left(\hat{c}_{i,n}(x)\right), d\right], \tag{9}$$

where $\text{SDF}_n$ denotes the Signed Distance Function for obstacle $i$ in frame $n$, which can change across frames in the case of moving obstacles. While this application is similar to the one proposed by GMD (Karunratanakul et al., 2023), ControlMM offers enhanced functionality by enabling obstacle avoidance for any joint at any frame, rather than being limited to the root trajectory (pelvis) as proposed in GMD.

**Body Part Timeline Control.** ControlMM supports motion generation conditioned on multiple joints, enabling control over body parts. To support multiple prompts corresponding to various body parts and timelines, ControlMM processes each prompt sequentially. Initially, it generates motion without any body part control, then iteratively refines the motion by incorporating prompts conditioned on the specified body parts and timeline constraints from the prior generation. Since ControlMM allows spatial control signals to target any joint and frame, partial body or temporal frame control is applicable within this framework. The detail of this process is described in A.11.

## 5 EXPERIMENT

**Datasets.** We conduct comprehensive experiments on the HumanML3D dataset (Guo et al., 2022b) HumanML3D covers a wide variety motions. It includes 14,616 motion sequences accompanied by 44,970 text descriptions. The textual data contains 5,371 unique words. The motion sequences are sourced from AMASS (Mahmood et al., 2019) and HumanAct12 (Guo et al., 2020).

**Evaluation.** We follow the evaluation protocol from OmniControl (Xie et al., 2023) which combines evaluation of quality from HumanML3D(Guo et al., 2022b) and trajectory error from GMD (Karunratanakul et al., 2023). The Frechet Inception Distance (FID) is used to assess the naturalness of the generated motion. R-Precision measures how well the generated motion aligns with its corresponding text prompt, while Diversity captures the variability within the generated motion. To assess control performance, we use the foot skating ratio, following Karunratanakul et al. (2023), as an indicator of coherence between the motion trajectory and the physical plausibility of the human motion. We also report Trajectory error, Location error, and Average error of the controlled joint positions in keyframes to evaluate control accuracy. All models are trained to generate 196 frames for evaluation, using 5 levels of sparsity in the control signal: 1, 2, 5, 49 (25% density), and 196 keyframes (100% density). Keyframes are sampled randomly, and we report the average performance across all density levels. During both training and evaluation, models receive ground-truth trajectories as spatial control signals.

### 5.1 QUANTITATIVE COMPARISON TO STATE-OF-THE-ART APPROACHES

GMD (Karunratanakul et al., 2023) only addresses the pelvis location on the ground plane (xz coordinates). To ensure a fair comparison, we follow OmniControl (Xie et al., 2023) and compare GMD in managing the full 3D position of the pelvis (xyz coordinates). The first section of Table 1 resents results for models trained on the pelvis alone to ensure a fair comparison with previous state-of-the-art methods on the HumanML3D (Guo et al., 2022b) dataset. → means closer to real data is better. Our model demonstrates significant improvements across all evaluation metrics. When compare to TLControl, the FID score notably decreased from 0.271 to 0.061, the R-Precision increased from 0.779 to 0.809, indicating superior generation quality. In terms of spatial control accuracy, both Trajectory Error and Location Error dropped to zero, while the average error decreased to 0.91 cm, indicating highly precise spatial control. Furthermore, our model outperforms existing methods in both Diversity and Foot Skating Ratio metrics. In the second section, *Train on All Joints*, we follow the evaluation from OmniControl (Xie et al., 2023), as our model supports control of any joint, not just the root (pelvis). We train the model to control multiple joints, specifically the pelvis, left foot, right foot, head, left wrist, and right wrist. The *Cross* experiment shows 63 cross-joint combinations (details in Appendix. A.13), while *Average* reflects the average performance across each joint. Our model outperforms all other methods across all joint configurations, including *Average* and *Cross*. Compared to OmniControl, our model delivers superior quality in *Cross*, evidenced by a FID score drop to 0.049 and an R-Precision increase to 0.811. In contrast, OmniControl struggles with multiple joints, as its FID score spikes to 0.624—almost triple its performance on the pelvis alone. Moreover, our model maintains zero Trajectory and Location Errors, while preserving Diversity, whereas OmniControl's *Trajectory Error* increase to 0.2147 and Diversity significantly drops to 9.016, indicating our model's robust handling of multiple control signals.

### 5.2 QUALITATIVE COMPARISON TO STATE-OF-THE-ART APPROACHES

We visualize the generated motion using **GMD** (Karunratanakul et al., 2023) and **OmniControl** (Xie et al., 2023) in Fig. 5. The motion is generated based on the prompt "a person walks forward and waves his hands," with the pelvis and right wrist controlled in a zigzag pattern. Since **GMD** can only control the pelvis, we apply control only to the pelvis for GMD. However, it fails to follow the zigzag pattern, tending instead to move in a straight line. **OmniControl** receives control signals for both the pelvis and right wrist. Yet, it not only fails to follow the root trajectory (pelvis) but also does not adhere to the zigzag pattern for the right wrist. In contrast, our **ControlMM** demonstrates realistic motion with precise spatial control for both the pelvis and the right wrist, accurately following the intended zigzag pattern.

Table 1: Comparison of text-condition motion generation with spacial control signal on the HumanML3D. The first section, "Train on Pelvis Only," evaluates our model that was trained solely on the pelvis. The last section, "Train on All Joints", is trained on all joints and assessing performance for each one. The cross-section reports performance across various combinations of joints.

| Method | Joint | R-Precision Top-3 ↑ | FID ↓ | Diversity → | Foot Skating Ratio ↓ | Traj. Err. (50 cm) ↓ | Loc. Err. (50 cm) ↓ | Avg. Err. ↓ |
|---|---|---|---|---|---|---|---|---|
| Real | - | 0.797 | 0.002 | 9.503 | - | 0.0000 | 0.0000 | 0.0000 |
| **Train on Pelvis Only** | | | | | | | | |
| MDM | | 0.602 | 0.698 | 9.197 | 0.1019 | 0.4022 | 0.3076 | 0.5959 |
| PriorMDM | | 0.583 | 0.475 | 9.156 | 0.0897 | 0.3457 | 0.2132 | 0.4417 |
| GMD | | 0.665 | 0.576 | 9.206 | 0.1009 | 0.0931 | 0.0321 | 0.1439 |
| OmniControl (on pelvis) | Pelvis | 0.687 | 0.218 | 9.422 | 0.0547 | 0.0387 | 0.0096 | 0.0338 |
| TLControl | | 0.779 | 0.271 | 9.569 | - | 0.0000 | 0.0000 | 0.0108 |
| MotionLCM | | 0.752 | 0.531 | 9.253 | - | 0.1887 | 0.0769 | 0.1897 |
| **ControlMM** (on pelvis) | | **0.809** | **0.061** | **9.496** | **0.0547** | **0.0000** | **0.0000** | **0.0098** |
| **Train on All Joints** | | | | | | | | |
| OmniControl | | 0.691 | 0.322 | **9.545** | 0.0571 | 0.0404 | 0.0085 | 0.0367 |
| TLControl | | 0.779 | 0.271 | 9.569 | - | 0.0000 | 0.0000 | **0.0108** |
| **ControlMM** | | **0.804** | **0.071** | 9.453 | **0.0546** | **0.0000** | **0.0000** | 0.0127 |
| OmniControl | | 0.696 | 0.280 | **9.553** | 0.0692 | 0.0594 | 0.0094 | 0.0314 |
| TLControl | Left Foot | 0.768 | 0.368 | 9.774 | - | 0.0000 | 0.0000 | 0.0114 |
| **ControlMM** | | **0.804** | **0.076** | 9.389 | **0.0559** | **0.0000** | **0.0000** | **0.0072** |
| OmniControl | | 0.701 | 0.319 | **9.481** | 0.0668 | 0.0666 | 0.0120 | 0.0334 |
| TLControl | Right Foot | 0.775 | 0.361 | 9.778 | - | 0.0000 | 0.0000 | 0.0116 |
| **ControlMM** | | **0.805** | **0.074** | 9.400 | **0.0549** | **0.0000** | **0.0000** | **0.0068** |
| OmniControl | | 0.696 | 0.335 | **9.480** | 0.0556 | 0.0422 | 0.0079 | 0.0349 |
| TLControl | Head | 0.778 | 0.279 | 9.606 | - | 0.0000 | 0.0000 | 0.0110 |
| **ControlMM** | | **0.805** | **0.085** | 9.415 | **0.0538** | **0.0000** | **0.0000** | **0.0071** |
| OmniControl | | 0.680 | 0.304 | **9.436** | 0.0562 | 0.0801 | 0.0134 | 0.0529 |
| TLControl | Left Wrist | 0.789 | 0.135 | 9.757 | - | 0.0000 | 0.0000 | 0.0108 |
| **ControlMM** | | **0.807** | **0.093** | 9.374 | **0.0541** | **0.0000** | **0.0000** | **0.0051** |
| OmniControl | | 0.692 | 0.299 | **9.519** | 0.0601 | 0.0813 | 0.0127 | 0.0519 |
| TLControl | Right Wrist | 0.787 | 0.137 | 9.734 | - | 0.0000 | 0.0000 | 0.0109 |
| **ControlMM** | | **0.805** | **0.099** | 9.340 | **0.0539** | **0.0000** | **0.0000** | **0.0050** |
| OmniControl | Average | 0.693 | 0.310 | **9.502** | 0.0608 | 0.0617 | 0.0107 | 0.0404 |
| **ControlMM** | | **0.805** | **0.083** | 9.395 | **0.0545** | **0.0000** | **0.0000** | **0.0072** |
| OmniControl | Cross | 0.672 | 0.624 | 9.016 | 0.0874 | 0.2147 | 0.0265 | 0.0766 |
| **ControlMM** | | **0.811** | **0.049** | **9.533** | **0.0545** | **0.0000** | **0.0000** | **0.0126** |

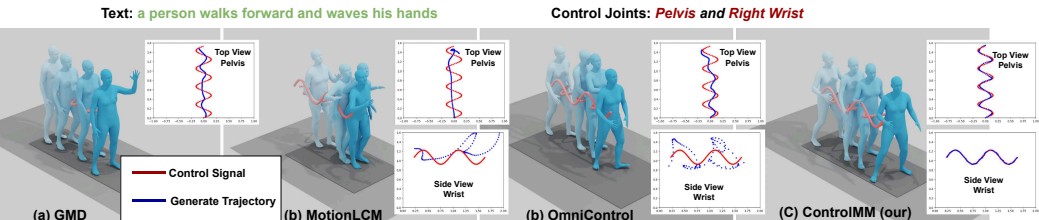

Figure 5: Visualization comparisons to state-of-the-art methods. The plots on the top display the top view of pelvis control (root trajectory), while the bottom plot shows the side view of the right wrist. Red represents the control signal, and Blue represents the generated joint motion.

## 5.3 BODY PART EDITING

With spatial signal control, our model is capable of conditioning on multiple joints, which can be treated as distinct body parts, while generating the remaining body parts based on text input. In Table 2 We quantitatively compare our approach to existing methods designed for this task, including MDM (Tevet et al., 2022b) and MMM (Pinyoanuntapong et al., 2024b). Additionally, we compare

it with OmniControl (Xie et al., 2023), which also supports spatial signal control. However, our evaluation demonstrates that OmniControl performs poorly in this task. Following the evaluation protocol from (Pinyoanuntapong et al., 2024b), we condition the lower body parts on ground truth for all frames and generate the upper body based on text descriptions using the HumanML3D dataset (Guo et al., 2022b). Our model is evaluated without retraining, using the same model as in the *Train on All Joints* setup, ensuring a fair comparison with OmniControl, which is trained on a subset of joints. Specifically, we condition only on the pelvis, left foot, and right foot as the lower body signals.

The results show that MDM struggles significantly when conditioned on multiple joints, with the FID score increasing to 4.827. Although OmniControl supports multiple joint control, our experiments reveal that it also suffers under these conditions, with its FID score rising to 1.213. This is consistent with the Cross-Joint evaluation in Table 1, which evaluate on multiple joint combination, where OmniControl's FID score deteriorates considerably. MMM performs well in this task but requires retraining with separate codebooks for upper and lower body parts. In contrast, our model outperforms all other methods across all metrics without any retraining. When comparing to the 'Train on Pelvis Only' setup in Table 1, our model achieves similar FID and R-Precision scores, highlighting its robustness in handling multiple joint control signals.

Table 2: Quantitative result of upper body editing task on HumanML3D dataset.

| Method | R-precision ↑ | | | FID ↓ | MM-Dist ↓ | Diversity → |
|---|---|---|---|---|---|---|
| | Top1 | Top2 | Top3 | | | |
| MDM (Tevet et al., 2022b) | 0.298 | 0.462 | 0.571 | 4.827 | 4.598 | 7.010 |
| OmniControl (Xie et al., 2023) | 0.374 | 0.550 | 0.656 | 1.213 | 5.228 | 9.258 |
| MMM (Pinyoanuntapong et al., 2024b) | 0.500 | 0.694 | 0.798 | 0.103 | 2.972 | 9.254 |
| ControlMM (ours) | **0.517** | **0.708** | **0.804** | **0.074** | **2.945** | **9.380** |

## 6 ABLATION STUDY

### 6.1 QUALITATIVE RESULTS

We visualize each component in Fig. 6 by controlling the pelvis and left wrist with the text prompt "a person walks in a circle counter-clockwise." (a) **Motion Control Model**: The overall motion is realistic but the controlled joints (pelvis and left wrist) deviate significantly from the spatial control signals. (b) **Logits Editing**: The root positions (pelvis) are closer to the spatial control signal, but the left wrist positions remain inaccurate. (c) **Codebook Editing**: Both the pelvis and left wrist positions align more closely with the spatial control signals, but the motion lacks realism because Codebook Editing only adjusts the motion at the end of the generation process. (d) **Full Model**: With all components active, the model generates realistic motion with high precision to match the control signals, while OmniControl fails to follow the control signals for both the pelvis and left wrist.

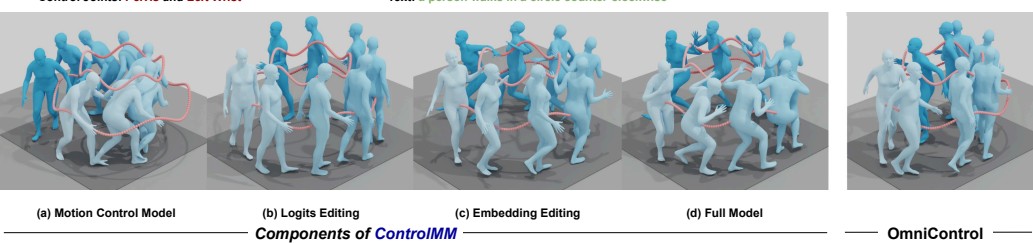

Figure 6: Qualitative comparisons of each component and the baseline

### 6.2 COMPONENT ANALYSIS

The key components of our model are *Logits Editing*, *Codebook Editing*, and *Motion Control Model*. To understand how each component impact the quality and spatial control error. We conduct ablation experiments using same evaluation as Table 1.

Table 3: Ablation results of all combinations of the main components.

| # | Logits Editing. | Codebook Editing. | Motion Control Model | R-Precision Top-3 ↑ | FID ↓ | Diversity → | Foot Skating Ratio ↓ | Traj. Err. (50 cm) ↓ | Loc. Err. (50 cm) ↓ | Avg. Err. ↓ |
|---|---|---|---|---|---|---|---|---|---|---|
| 1 | ✗ | ✗ | ✗ | 0.807 | 0.095 | **9.672** | **0.0527** | 0.5066 | 0.3511 | 0.6318 |
| 2 | ✓ | ✗ | ✗ | 0.813 | 0.105 | 9.615 | 0.0529 | 0.2323 | 0.1175 | 0.2361 |
| 3 | ✗ | ✓ | ✗ | 0.786 | 0.190 | 9.294 | 0.0616 | 0.0063 | 0.0005 | 0.0283 |
| 4 | ✓ | ✓ | ✗ | 0.795 | 0.142 | 9.402 | 0.0577 | 0.0032 | 0.0002 | 0.0218 |
| 5 | ✗ | ✗ | ✓ | 0.802 | 0.128 | 9.475 | 0.0594 | 0.3914 | 0.2400 | 0.4041 |
| 6 | ✓ | ✗ | ✓ | **0.814** | **0.051** | 9.557 | 0.0541 | 0.1302 | 0.0623 | 0.1660 |
| 7 | ✗ | ✓ | ✓ | 0.806 | 0.069 | 9.425 | 0.0568 | 0.0005 | 0.0000 | 0.0124 |
| 8 | ✓ | ✓ | ✓ | 0.809 | 0.061 | 9.496 | 0.0547 | **0.0000** | **0.0000** | **0.0098** |

From Table 3, without any control (#1), the model achieves the highest diversity and the lowest Foot Skating Ratio, indicating strong realism in the generated motion. The FID score is also on par. However, all spatial errors are poor due to the absence of spatial control components in the model. For the experiments without *Codebook Editing* (#1, #2, #5, #6), both FID scores and R-Precision are notable, particularly in #6, which combines *Logits Editing* and the *Motion Control Model* to enhance generation quality. In contrast, #3, which solely uses *Codebook Editing*, exhibits the worst FID score and Foot Skating Ratio while showing acceptable spatial control errors. This experiment highlights that while *Codebook Editing* can reduce generation errors, it may negatively impact the overall quality. Conversely, incorporating *Logit Editing* and *Motion Control Model* during each iteration improves both quality and spatial control errors, as demonstrated in #8.

## 6.3 DENSITY OF SPATIAL CONTROL SIGNAL

Table 4: Ablation results on different densities.

| Density | R-Precision Top-3 ↑ | FID ↓ | Diversity → | Foot Skating Ratio ↓ | Traj. Err. (50 cm) ↓ | Loc. Err. (50 cm) ↓ | Avg. Err. ↓ |
|---|---|---|---|---|---|---|---|
| 1 | 0.804 | 0.077 | 9.526 | 0.0551 | 0.0000 | 0.0000 | 0.0010 |
| 2 | 0.806 | 0.087 | 9.475 | 0.0553 | 0.0000 | 0.0000 | 0.0034 |
| 5 | 0.811 | 0.078 | 9.499 | 0.0553 | 0.0000 | 0.0000 | 0.0098 |
| 49 (50%) | 0.812 | 0.055 | 9.507 | 0.0536 | 0.0001 | 0.0000 | 0.0168 |
| 196 (100%) | 0.814 | 0.054 | 9.514 | 0.0543 | 0.0002 | 0.0000 | 0.0164 |

In table 4, we provide a detailed analysis of ControlMM's performance across five different spatial control density levels, where the model is trained for pelvis control using the HumanML3D dataset. The results show that increasing the spatial control improves the quality: the FID score decreases from 0.077 with 1-frame control to 0.054 with full 196-frame (100%) control. Similarly, R-Precision improves from 0.804 at 1-frame density to 0.814 at 196-frame (100%) density. However, the Average Error shows the opposite trend—more spatial control leads to higher error, as the model is required to target more specific points.

## 7 CONCLUSION

In this work, we present ControlMM, a new method that incorporates spatial control signals into the Masked Motion Model. ControlMM is the first model that enables precise control over quantized motion tokens while maintaining high-quality motion generation at faster speeds, consistently outperforming diffusion-based controllable frameworks. ControlMM introduces two key innovations: *Masked Consistency Modeling* uses random masking and reconstruction to ensure that the generated motions are of high fidelity, while also reducing inconsistencies between the input control signals and the motions produced. *Inference-Time Logit and Codebook Editing* fine-tunes the predicted motion distribution to better match the input control signals, enhancing precision and making ControlMM adaptable for various tasks. ControlMM has a wide range of applications, including any joint any frame control, obstacle avoidance, and body part timeline control.

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

# A APPENDIX

## A.1 OVERVIEW

The supplementary material is organized into the following sections:

- Section A.2: Pseudo Code of ControlMM Inference
- Section A.3: Implementation Details
- Section A.4: Inference speed, quality, and errors Details
- Section A.5: Speed of each component
- Section A.6: Quantitative result for all joints of ControlMM-Fast
- Section A.7: Ablation on less number of generation steps
- Section A.8: Analysis of *Logits Editing* and *Motion Control Model*
- Section A.9: The challenges of Motion Control Model
- Section A.10: Dual-Space Categorical Straight-Through Estimator
- Section A.11: Body Part Timeline Control
- Section A.12: KIT Dataset
- Section A.13: Cross Combination

Video visualization can be found at `https://anonymous-ai-agent.github.io/CAM`

## A.2 PSEUDO CODE OF CONTROLMM INFERENCE

---

**Algorithm 1** ControlMM Inference

---

**Require:** Masked Motion Model ($MMM$), Motion Control Model ($MCM$), mask scheduling function $\gamma(\cdot)$, spatial control signals $s$ (if any), text prompts $W$ (if any).

1: $X_{\overline{M}} \leftarrow [Mask]$        ▷ Start with all mask tokens
2: **for all** $t$ from 1 to $T$ **do**        ▷ Unmask process in $T$ steps
3:      $\{\boldsymbol{f}\} \leftarrow MCM(X_{\overline{M}}, W, s; \phi)$        ▷ **Motion Control Model**
4:      $l \leftarrow MMM(X_{\overline{M}}, \boldsymbol{p}, \{\boldsymbol{f}\}; \theta)$        ▷ **Masked Motion Model**
5:      **for all** $i$ from 1 to $I_l$ **do**        ▷ **Logits Editing**
6:          $l_{i+1} = l_i - \eta \nabla_{l_i} L_s(l_i, s)$
7:      **end for**
8:      $X_{\overline{M}} \leftarrow \gamma(l, t)$        ▷ mask out tokens based on logits $l$ at time step $t$
9: **end for**
10: **for all** $i$ from 1 to $I_e$ **do**        ▷ **Embedding Editing**
11:      $e_c^{i+1} = e_c^i - \eta \nabla_{e_c^i} L_s(e_c^i, s)$
12: **end for**
13: **return** $Decoder(e_c)$

---

## A.3 IMPLEMENTATION DETAILS

We modified the MoMask (Guo et al., 2023) model by retraining it with a cross-entropy loss applied to all tokens, instead of just the masked positions. This retrained model serves as our pretrained base model, and we kept the default hyperparameter settings unchanged. To improve robustness to text variation, we randomly drop 10% of the text conditioning, which also allows the model to be used for Classifier-Free Guidance (CFG). The weight for Eq. 5 is set to $\alpha = 0.1$. We use a codebook of size 512, with embeddings of size 512 and 6 residual layers. The Transformer embedding size is set to 384, with 6 attention heads, each with an embedding size of 64, distributed across 8 layers. This configuration demonstrates the feasibility of converting between two different embedding sizes and spaces using the Dual-Space Categorical Straight-Through Estimator. The encoder and decoder downsample the motion sequence length by a factor of 4 when mapping to token space. The learning rate follows a linear warm-up schedule, reaching 2e-4 after 2000 iterations, using AdamW optimization. The mini-batch size is set to 512 for training RVQ-VAE and 64 for training the Transformers. During inference, the CFG scale is set to $cfg = 4$ for the base layer and $cfg = 5$ for the

6 layers of residual, with 10 steps for generation. We use pretrained CLIP model (Radford et al., 2021) to generate text embeddings, which have a size of 512. These embeddings are then projected down to a size of 384 to match the token size used by the Transformer. *Motion Control Model* is a trainable copy of Masked Transformer with the zero linear layer connect to the output each layer of the Masked Transformer. During inference, *Logits and Codebook Editing* applies L2 loss with a learning rate of 0.06 for 100 iterations in *Codebook Editing* for each of the 10 generation steps and 600 iterations in *Logits Editing*. We apply temperature of 1 for all 10 steps and 1e-8 for residual layers. We follow the implementation from Karunratanakul et al. (2023); Xie et al. (2023); Wan et al. (2023), applying the spatial control signal only to joint positions and omitting rotations.

## A.4 INFERENCE SPEED, QUALITY, AND ERRORS

We compare the speed of three different configurations of our model against state-of-the-art methods as shown in Table 5. The first setting, **ControlMM-Fast**, uses 100 iterations of *Codebook Editing* without *Logits Editing*. This setup achieves results comparable to OmniControl, but is over 20 times faster. It also slightly improves the Trajectory and Location Errors, while the FID score is only 25% of OmniControl's, indicating high generation quality. The second setting, **ControlMM-Medium**, increases the *Codebook Editing* to 600 iterations, which further improves accuracy. The Location Error is reduced to zero, although the FID score slightly worsens. Lastly, the **ControlMM-Accurate** model, which is the default setting used in other tables in this paper, uses 600 iterations of *Codebook Editing* and 100 iterations of *Logits Editing*. This configuration achieves extremely high accuracy, with both the Trajectory and Location Errors reduced to zero and the Average Error below 1 cm (0.0098 meters). Importantly, these settings can be adjusted during inference without retraining the model, making them suitable for both real-time and high-performance applications.

Table 5: Comparison of Motion Generation Performance with Speed and Quality Metrics

| Model | Speed ↓ | R-Precision Top-3 ↑ | FID ↓ | Diversity → | Foot Skating Ratio ↓ | Traj. Err. (50 cm) ↓ | Loc. Err. (50 cm) ↓ | Avg. Err. ↓ |
|---|---|---|---|---|---|---|---|---|
| MDM | 10.14 s | 0.602 | 0.698 | 9.197 | 0.1019 | 0.4022 | 0.3076 | 0.5959 |
| PriorMDM | 18.11 s | 0.583 | 0.475 | 9.156 | 0.0897 | 0.3457 | 0.2132 | 0.4417 |
| GMD | 132.49 s | 0.665 | 0.576 | 9.206 | 0.1009 | 0.0931 | 0.0321 | 0.1439 |
| OmniControl | 87.33 s | 0.687 | 0.218 | 9.422 | 0.0547 | 0.0387 | 0.0096 | 0.0338 |
| ControlMM-Fast | 4.94 s | 0.808 | 0.059 | 9.444 | 0.0570 | 0.0200 | 0.0075 | 0.0550 |
| ControlMM-Medium | 25.23 s | 0.806 | 0.069 | 9.425 | 0.0568 | 0.0005 | 0.0000 | 0.0124 |
| ControlMM-Accurate | 71.72 s | 0.809 | 0.061 | 9.496 | 0.0547 | 0.0000 | 0.0000 | 0.0098 |

## A.5 SPEED OF EACH COMPONENT

We report the inference time for each component in Table 6, with all measurements taken on an NVIDIA A100. The **Base** model, which includes only the Masked Transformer with Residual layers and Decoder (without any spatial control signal module), has an inference time of 0.35 second. The **Motion Control Model** is highly efficient, requiring only 0.24 seconds for inference. The **Codebook Editing** and **Logits Editing** components take 24.65 seconds and 46.5 seconds, respectively. In total, the **ControlMM-Accurate** model has a generation time of 71.73 seconds. Note that this setting is using 100 iterations of **Codebook Editing** for 10 steps and 600 iterations of **Logits Editing**.

Table 6: Inference time of each component

| | Base | Motion Control Model | Codebook Editing | Logits Editing | Full |
|---|---|---|---|---|---|
| Speed in Seconds | 0.35 | 0.24 | 24.65 | 46.5 | 71.73 |

## A.6 QUANTITATIVE RESULT FOR ALL JOINTS OF CONTROLMM-FAST

Table 7 presents the evaluation results for ControlMM-Fast, which uses 100 iterations of **Codebook Editing** without **Logits Editing**. This evaluation includes a "cross" assessment that evaluates combinations of different joints, as detailed in Section A.13. The results can be compared to those of

the full model (ControlMM-Accurate) and state-of-the-art models shown in Table 1. Additionally, "lower body" refers to the conditions involving the left foot, right foot, and pelvis, which allows for the evaluation of upper body editing tasks, as illustrated in Table 2.

Table 7: Quantitative result for all joints of ControlMM-Fast

| Joint | R-Precision Top-3 ↑ | FID ↓ | Diversity ↑ | Foot Skating Ratio ↓ | Traj. Err. (50 cm) ↓ | Loc. Err. (50 cm) ↓ | Avg. Err. ↓ |
|---|---|---|---|---|---|---|---|
| pelvis | 0.806 | 0.067 | 9.453 | 0.0552 | 0.0446 | 0.0151 | 0.0691 |
| left foot | 0.806 | 0.074 | 9.450 | 0.0561 | 0.0495 | 0.0105 | 0.0484 |
| right foot | 0.808 | 0.069 | 9.416 | 0.0566 | 0.0453 | 0.0099 | 0.0469 |
| head | 0.810 | 0.080 | 9.411 | 0.0555 | 0.0525 | 0.0148 | 0.0665 |
| left wrist | 0.809 | 0.085 | 9.380 | 0.0545 | 0.0467 | 0.0108 | 0.0534 |
| right wrist | 0.807 | 0.095 | 9.387 | 0.0549 | 0.0498 | 0.0113 | 0.0538 |
| Average | 0.808 | 0.079 | 9.416 | 0.0555 | 0.0481 | 0.0121 | 0.0563 |
| cross | 0.812 | 0.050 | 9.515 | 0.0545 | 0.0330 | 0.0101 | 0.0739 |
| lower body | 0.807 | 0.084 | 9.396 | 0.0491 | 0.0312 | 0.0050 | 0.0633 |

### A.7 ABLATION ON LESS NUMBER OF GENERATION STEP

In this section, we perform an ablation study on the number of steps used in the generation process. Following the MoMask architecture (Guo et al., 2023), we adopt the same setting of 10 steps for generation. However, the integration of *Logits Editing* and the *Motion Control Model* enhances the quality of the generated outputs with fewer steps, as demonstrated in Table 8. Notably, with just 1 step, the results are already comparable to those achieved by TLControl (Wan et al., 2023). Furthermore, after 4 steps, the evaluation metrics are on par with those obtained after 10 steps.

Table 8: Quantitative result for different number of steps with *Logits Editing* and *Motion Control Model*

| # of steps | R-Precision Top-3 ↑ | FID ↓ | Diversity → | Foot Skating Ratio ↓ | Traj. Err. (50 cm) ↓ | Loc. Err. (50 cm) ↓ | Avg. Err. ↓ |
|---|---|---|---|---|---|---|---|
| 1 | 0.779 | 0.276 | 9.353 | 0.0545 | 0.0002 | 0.0000 | 0.0110 |
| 2 | 0.792 | 0.118 | 9.436 | 0.0530 | 0.0001 | 0.0000 | 0.0100 |
| 4 | 0.806 | 0.068 | 9.468 | 0.0543 | 0.0001 | 0.0000 | 0.0098 |
| 6 | 0.809 | 0.063 | 9.478 | 0.0545 | 0.0001 | 0.0000 | 0.0098 |
| 8 | 0.810 | 0.059 | 9.511 | 0.0543 | 0.0001 | 0.0000 | 0.0098 |
| 10 | 0.809 | 0.061 | 9.496 | 0.0547 | 0.0000 | 0.0000 | 0.0098 |

To further investigate the influence of *Logits Editing* and the *Motion Control Model* for lesser steps, we remove these components and experiment with various numbers of steps, as shown in Table 9. Reducing the number of steps significantly decreases the quality of the generated outputs, resulting in an FID score of 1.196 with only 1 step. Even with 10 steps, the FID score remains at 0.190, highlighting the improvements by integrating *Logits Editing* and the *Motion Control Model*.

Table 9: Quantitative result for different number of steps without *Logits Editing* and *Motion Control Model*

| # of steps | R-Precision Top-3 ↑ | FID ↓ | Diversity → | Foot Skating Ratio ↓ | Traj. Err. (50 cm) ↓ | Loc. Err. (50 cm) ↓ | Avg. Err. ↓ |
|---|---|---|---|---|---|---|---|
| 1 | 0.716 | 1.196 | 8.831 | 0.0715 | 0.0070 | 0.0006 | 0.0271 |
| 2 | 0.758 | 0.462 | 9.182 | 0.0672 | 0.0067 | 0.0005 | 0.0276 |
| 4 | 0.782 | 0.238 | 9.236 | 0.0628 | 0.0066 | 0.0005 | 0.0281 |
| 6 | 0.787 | 0.203 | 9.276 | 0.0614 | 0.0061 | 0.0005 | 0.0282 |
| 8 | 0.787 | 0.193 | 9.272 | 0.0613 | 0.0062 | 0.0005 | 0.0283 |
| 10 | 0.786 | 0.190 | 9.294 | 0.0616 | 0.0063 | 0.0005 | 0.0283 |

## A.8 ANALYSIS OF *Logits Editing* AND *Motion Control Model*

To understand the impact of *Logits Editing* and *Motion Control Model* on the generation process, we visualize the maximum probability for each token prediction from the Masked Transformer. The model predicts 49 tokens over 10 steps. We show results both before and after applying *Logits Editing*, and with and without the *Motion Control Model*. The maximum probability can be expressed as the relative value of the logits corresponding to all codes in the codebook in the specific token position and step, as computed by the Softmax function. We visualize the output using the Softmax function instead of Gumbel-Softmax. By removing the Gumbel noise, Gumbel-Softmax reduces to a regular Softmax function:

$$p_i = \frac{\exp(\ell_i)}{\sum_{j=1}^{k} \exp(\ell_j)}$$

The generation is conditioned by the text prompt, "a person walks in a circle counter-clockwise" with control over the pelvis and right hand throughout the entire trajectory. In the plot, darker blue colors represent lower probabilities (0), while yellow represents higher probabilities (1).

**Without *Motion Control Model***

In the first step (step 0), the probability is low but increases significantly in the subsequent steps. After applying *Logits Editing*, the probability improves slightly, as shown in Fig. 8 and 7. Eventually, the probability saturates in the later steps (see Figure 11). Since the probability of most token predictions approaches one, *Logits Editing* cannot further modify the logits, preventing any updates to the trajectory.

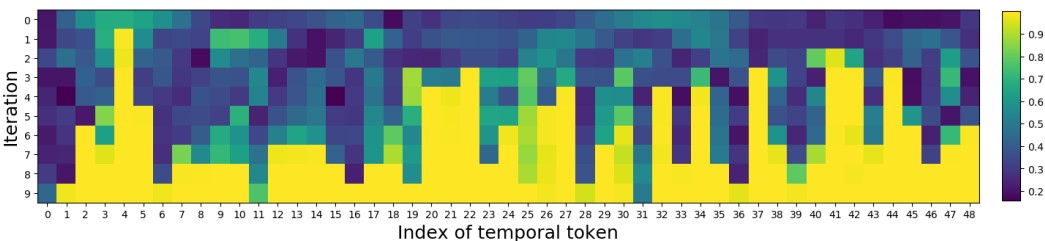

Figure 7: The maximum probability of the each token **without** *Motion Control Model* **before** *Logits Editing* of each all 49 tokens and 10 steps.

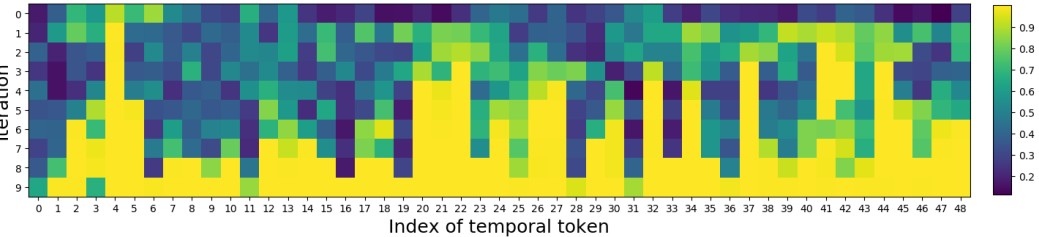

Figure 8: The maximum probability of the each token **without** *Motion Control Model* **after** *Logits Editing* of each all 49 tokens and 10 steps.

**With *Motion Control Model***

With the introduction of the *Motion Control Model*, the probability of token predictions is significantly higher in the initial step compared to the scenario without the *Motion Control Model*, as illustrated in Figures 9 and 10. Moreover, the maximum probability does not saturate to one, indicating that there is still room to adjust the logits for trajectory editing.

This enhancement leads to improved generation quality within fewer steps, as detailed in Section A.7. Notably, just 4 steps using the *Motion Control Model* yield a quality comparable to that

achieved in 10 steps without it, where the latter still exhibits suboptimal quality and high average error.

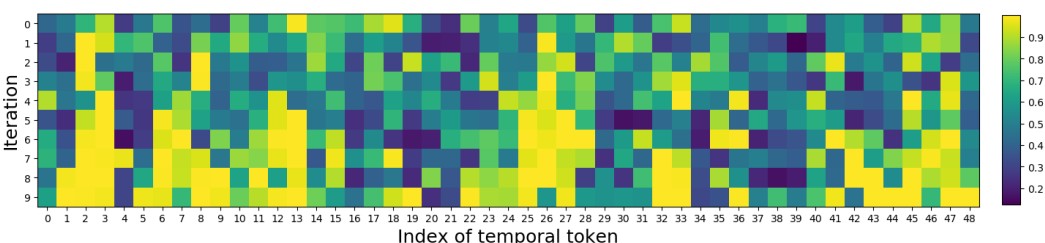

Figure 9: The maximum probability of the each token **with** *Motion Control Model* **before** *Logits Editing* of each all 49 tokens and 10 steps.

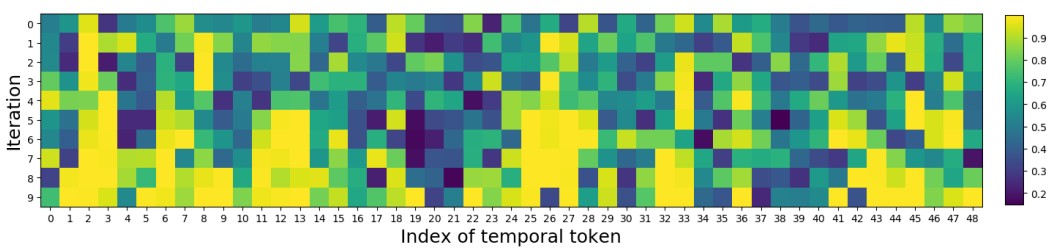

Figure 10: The maximum probability of the each token **with** *Motion Control Model* **after** *Logits Editing* of each all 49 tokens and 10 steps.

**Average of maximum probability of all tokens in each step** To clearly illustrate the increasing probability or confidence of the model predictions across all 10 steps, as shown in Fig. 11. In this figure, the blue line represents the average probability of token predictions **With the *Motion Control Model***, while the red line denotes the average probability **Without the *Motion Control Model***. The solid line indicates the average probability prior to the application of *Logits Editing*. This shows that the probability increases significantly in the very first step for the **With the *Motion Control Model***.

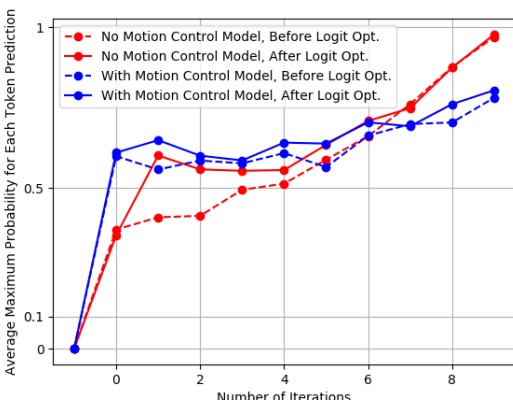

Figure 11: Average Maximum Probability for Each Token Prediction

## A.9 THE CHALLENGES OF MOTION CONTROL MODEL

**Ambiguity of Motion Control Signal**

Unlike adding conditional control to text-to-image models, where the control signal can directly insert values at the pixel to control and set '0' at pixels with no control. However, motion control

introduces ambiguity, both a control signal at the origin and no control can be represented as '0'. To address this, the relative difference between the generated motion at the current step and the absolute control signal is calculated and concatenated with the control signal to resolve the ambiguity as shown in 12.

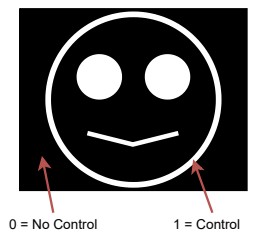 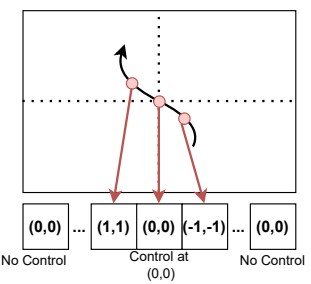 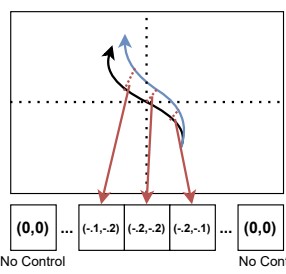

(a) Image Control      (b) Absolute Motion Control      (b) Relative Motion Control

Figure 12: The difference between control signals: **(a) Image Control**: 0 means no control, 1 means control. **(b) Absolute Motion Control**: ambiguous between control signal at origin and no control. **(c) Relative Motion Control**: no ambiguity. Black curve: spatial control signal. Blue curve: decoded spatial signal from generated motion

**Approximated Mask Embedding for Decoder**

As discussed above, motion Control Model requires the spatial signal difference as model input to avoid control signal ambiguity. To obtain the spatial signal difference, the model needs to decode [Mask] tokens for an initial motion token generation. The generated motion is compared with the control signal to obtain the spatial signal difference. However, [Mask] tokens are only used for the Masked Transformer, and there is no [Mask] token in the codebook, making it impossible to reconstruct motion from Masked Transformer embeddings. To address this issue, we approximate the [Mask] token for the codebook space by the average of all codebook. We visualize the embedding of the **[Mask] token (black)** compared to all **Transformer tokens (red)**, as shown in Fig. 13. The visualization indicates that the [Mask] embedding is approximately the average of all embeddings. By using the average of all embeddings for the [Mask] position, we can utilize the relative differences between the generated motion and the control motion for **Motion Control Model**.

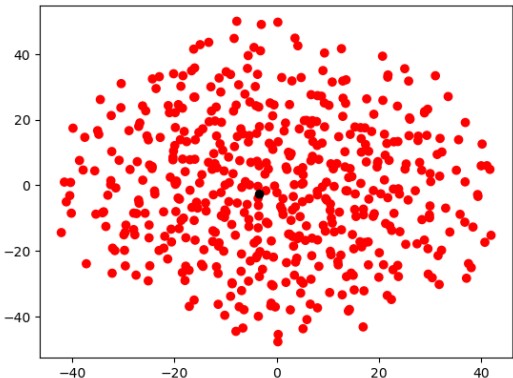

Figure 13: t-SNE visualization of the embeddings for all **Transformer tokens (red)**, comparing to the **[Mask] token (black)**.

A.10   DUAL-SPACE CATEGORICAL STRAIGHT-THROUGH ESTIMATOR

In diffusion models, guided diffusion (Dhariwal & Nichol, 2021) applies classifier guidance on diffusion noise, we adapt the concept for Masked Motion Model. However, applying guidance directly

to embeddings is impractical for Masked Models, as their Masked Transformers use learnable tokens that differ from the codebook space which requires for decoder of the Motion Tokenizer to reconstruct motion tokens to raw motion space. Instead, we propose **Logits Editing**, directly optimizing the logits which can approximate both the codebook space and Masked Transformers learnable token space.

To reconstruct motion from Transformer tokens, the tokens must first be mapped to their corresponding codebook embeddings using the same indices before being fed into the decoder. However, this index-based lookup operation is inherently non-differentiable, which obstructs guidance from the generated motion through the gradient backpropagation.

*Dual-Space Categorical Straight-Through Estimator (DCSE)* performs weighted average sampling of the codebook $C$ w.r.t. the probability distribution $p$. Given the output logits $l$ from the Transformer, instead of using the non-differentiable $\arg\max$ operation to select embedding from the codebook, we apply the Gumbel-Softmax function (Jang et al., 2017) to obtain a probability distribution as a smooth differentiable approximation alternative to the $\arg\max$ operation, producing $k$-dimensional sample vectors $p$.

$$p_i = \frac{\exp\left((\ell_i + g_i)/\tau\right)}{\sum_{j=1}^{k} \exp\left(\ell_j/\tau\right)} \qquad (10)$$

where $\tau$ refers to temperature and $g$ represents Gumbel noise with $g_1, \ldots, g_k$ being independent and identically distributed (i.i.d.) samples from a Gumbel$(0, 1)$ distribution. The Gumbel$(0, 1)$ distribution can be sampled via inverse transform sampling by first drawing $u \sim \text{Uniform}(0, 1)$ and then computing $g = -\log(-\log(u))$.

From sample vectors $p$, the approximated embedding can be obtained from weighted sampling of Transformer token space $e_j$

$$e_t = \sum_{j=1}^{k} p_i \cdot e_j \qquad (11)$$

or from code $c_j$ in Codebook $C$.

$$e_c = \sum_{j=1}^{k} p_i \cdot c_j \qquad (12)$$

In our implementation, we adopt the configuration from MoMask (Guo et al., 2023), using a codebook embedding size of 512 and a Transformer token size of 384. With this setup, we demonstrate that conversion across different spaces is feasible, even when the embedding sizes differ, as long as both spaces refer to the same set of indices. This allows for flexible representation across latent spaces while maintaining consistency in how the embeddings are referenced.

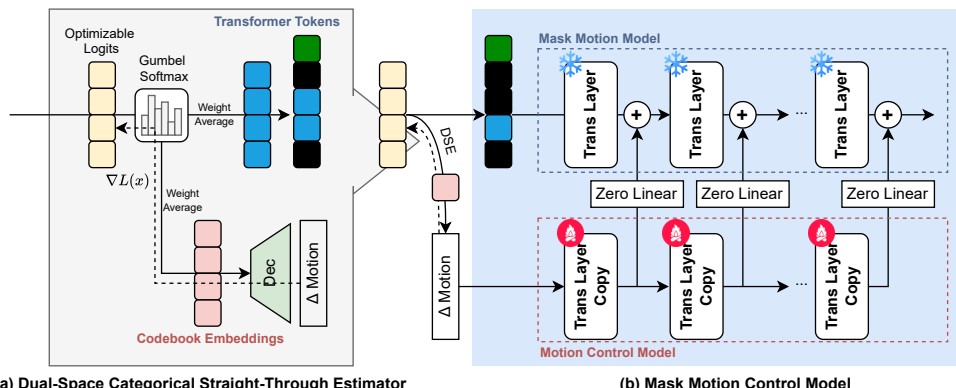

Figure 14: (a) Dual-Categorical Straight-Through Estimator (b) Mask Motion Control Model

### A.11 BODY PART TIMELINE CONTROL

Generating multiple body parts based on their respective text prompts is not straightforward, as the HumanML3D dataset provides only a single prompt for each motion without specific descriptions for individual body parts. However, our model can conditionally generate outputs based on spatial signals, which allows us to manipulate and control the generation process.

To achieve this, we first generate the entire body and motion for all frames. Next, we generate a new prompt related to the next body part, using the previously generated body parts as a condition. This process can be repeated multiple times to create motion for each body part based on its corresponding text prompt, as illustrated in Fig. 15.

It is important to note that this approach may lead to out-of-distribution generation since the model has not been trained on combinations of multiple body parts with their associated text prompts. However, our model handles out-of-distribution generation effectively due to the use of *Logits Editing* and *Codebook Editing*.

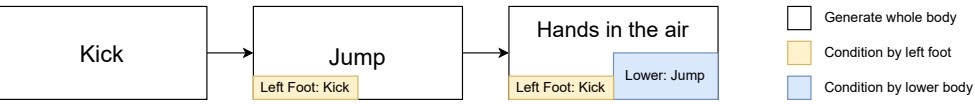

Figure 15: Process of generating body parts with multiple text inputs over a specific timeline

### A.12 KIT DATASET

We also tested ControlMM on the KIT dataset and compared it to state-of-the-art (SOTA) methods. Despite the KIT dataset being significantly smaller than HumanML3D, ControlMM consistently outperformed other SOTA methods in both quality and precise control, demonstrating its robustness.

Table 10: Comparison of text-condition motion generation with spatial control signal on the KIT .

| Method | R-Precision Top-3 ↑ | FID ↓ | Diversity → | Traj. Err. (50 cm) ↓ | Loc. Err. (50 cm) ↓ | Avg. Err. ↓ |
|---|---|---|---|---|---|---|
| PriorMDM | 0.397 | 0.851 | 10.518 | 0.3310 | 0.1400 | 0.2305 |
| GMD | 0.382 | 1.565 | 9.664 | 0.5443 | 0.3003 | 0.4070 |
| OmiControl | 0.397 | 0.702 | **10.927** | 0.1105 | 0.0337 | 0.0759 |
| TLControl | **0.757** | 0.432 | 10.723 | 0.0028 | 0.0011 | 0.0276 |
| ControlMM | 0.747 | **0.378** | 10.527 | **0.0018** | **0.0001** | **0.0160** |

## A.13 CROSS COMBINATION

We follow the evaluation *Cross Combination* from OmniControl (Xie et al., 2023), evaluating multiple combinations of joints as outlined in Table 1. A total of 63 combinations are randomly sampled during the evaluation process as follow.

1. pelvis
2. left foot
3. right foot
4. head
5. left wrist
6. right wrist
7. pelvis, left foot
8. pelvis, right foot
9. pelvis, head
10. pelvis, left wrist
11. pelvis, right wrist
12. left foot, right foot
13. left foot, head
14. left foot, left wrist
15. left foot, right wrist
16. right foot, head
17. right foot, left wrist
18. right foot, right wrist
19. head, left wrist
20. head, right wrist
21. left wrist, right wrist
22. pelvis, left foot, right foot
23. pelvis, left foot, head
24. pelvis, left foot, left wrist
25. pelvis, left foot, right wrist
26. pelvis, right foot, head
27. pelvis, right foot, left wrist
28. pelvis, right foot, right wrist
29. pelvis, head, left wrist
30. pelvis, head, right wrist
31. pelvis, left wrist, right wrist
32. left foot, right foot, head

33. left foot, right foot, left wrist
34. left foot, right foot, right wrist
35. left foot, head, left wrist
36. left foot, head, right wrist
37. left foot, left wrist, right wrist
38. right foot, head, left wrist
39. right foot, head, right wrist
40. right foot, left wrist, right wrist
41. head, left wrist, right wrist
42. pelvis, left foot, right foot, head
43. pelvis, left foot, right foot, left wrist
44. pelvis, left foot, right foot, right wrist
45. pelvis, left foot, head, left wrist
46. pelvis, left foot, head, right wrist
47. pelvis, left foot, left wrist, right wrist
48. pelvis, right foot, head, left wrist
49. pelvis, right foot, head, right wrist
50. pelvis, right foot, left wrist, right wrist
51. pelvis, head, left wrist, right wrist
52. left foot, right foot, head, left wrist
53. left foot, right foot, head, right wrist
54. left foot, right foot, left wrist, right wrist
55. left foot, head, left wrist, right wrist
56. right foot, head, left wrist, right wrist
57. pelvis, left foot, right foot, head, left wrist
58. pelvis, left foot, right foot, head, right wrist
59. pelvis, left foot, right foot, left wrist, right wrist
60. pelvis, left foot, head, left wrist, right wrist
61. pelvis, right foot, head, left wrist, right wrist
62. left foot, right foot, head, left wrist, right wrist
63. pelvis, left foot, right foot, head, left wrist, right wrist

