# OpenReview forum: "ControlMM: Controllable Masked Motion Generation"
_ICLR.cc/2025/Conference — Submitted to ICLR 2025_

### Official Review · Reviewer_KEsH · 2024-10-22

**Soundness:** 2
**Presentation:** 1
**Contribution:** 2
**Rating:** 5
**Confidence:** 5

**Summary:**

This paper proposes a controllable motion generation model, which supports joint, obstacle, and timeline controls. The key technical contributions claimed in this paper are the mask consistency training and the latent optimization during inference. The experiments verify the effectiveness of the method.

**Strengths:**

The experiments verify the effectiveness of the method.

The videos are provided on the website.

**Weaknesses:**

The writing quality of this work should be significantly improved. For example, the citation format in L112 should be revised. The writing of the method is very hard to read, especially the notation system. For example, the notations $l$ in Eqn. 6 is not well specified.

Despite these, my main concern lies in the technical contribution. The method part seems to show something like "a bag of tricks". The key components are logit editing, codebook editing, and controlnet. The controlnet of controllable motion generation is first introduced by MotionLCM, which is not very novel to me. The ablation in Table 3 (L1 vs. L5) shows the control module performs not very well. This might be the reason for introducing the other two tricks. From the alation, it seems that embedding editing is the main component for editing precision (L1 vs L3). Therefore, the separate sub-components are not all effective enough.

I also have some concerns about the evaluation. The authors miss two latest baselines, TLControl and MotionLCM. The method requires inference optimization, which introduces more time. The efficiency is not well discussed in the main text. Besides, the generation quality is not reported in the main text, which is a simple baseline for comparison. Besides, whether the optimization in the latent space has some issues about the over-fitting or not is not well discussed. How are the parameters determined?

As shown in the demo website, some of the motions jitter a lot. Some of motions are not continuous. (e.g., "A figure walks forward in a zig zag pattern", "person is walking as if injured", "a person walks forward carrying something", "A person walks forward, casually greeting others with a wave or hello") Is this owing to the latent optimization of the method?

If my concerns can be resolved, I will update rating accordingly.

**Questions:**

see weakness

---

> ### Author Response · Authors · 2024-11-22
> **______(1) Response to Reviewer KEsH______**
>
> Thank you for your time and valuable feedback. We fixed the writing and notation in the updated manuscript. Below, we address your questions and comments:
> ```
> Q1 "a bag of tricks". Controlnet of controllable motion generation similar to MotionLCM, not very novel.
> ```
> **A1**: The fundamental problem of current controllable motion generation models is that they cannot simultaneously achieve high-quality motion generation and high-precision spatial control.  Compared to MotionLCM, ControlMM has significantly higher quality (FID 0.061 vs 0.531) and more precise (Avg Error 0.0098 vs 0.1897). The key difference is that current controllable models are based on diffusion models, which cannot fully capture the internet spatial and temporal dependencies in the complex human motion, thus leading to low generation quality (high FID score). On the other hand, the masked motion model, which relies on a BERT-like generative masked modeling, can lead to much improved generation quality by explicitly modeling and learning the joint distribution of motion tokens.  However, current masked motion models do not have spatial controllability. To address this challenge, we propose ControlMM that consists of two complementary components: motion control model and logits-codebook editing.
>
> **Motion Control Model**, inspired by ControlNet, aims to add spatial control signals into the pretrained masked motion model. However, the motion control model is very unique from following perspectives. 1) Overall working principle: motion control model achieves spatial control by modifying the underlying motion token distribution. ControlNet realizes spatial control by alternating diffusion process. (2) Architecture: motion control model is a partial copy of the original masked motion model, where it does not possess the motion token classifier in the original model.  ControlNet has the identical architecture as the original diffusion model. Moreover, motion control model operates on the discrete motion tokens extracted from motion tokenizer. (3) Training paradigm: motion control model is trained using the combination of (i) generative masking training,  (ii) consistency feedback  (iii) and training-time differential sampling. ControlNet is trained via the optimal denoising process.  (4) Inference process:  because the generative masking training learned motion token distribution conditioned on text and spatial control prompts, during inference, motion control model follows the parallel token decoding that iteratively sample from the learned distribution to  obtain the high-confident tokens, while remasking and repredicting low-confidence token in the following iterations.
>
> **Logits Editing + Codebook Editing** To the best of our knowledge, we are the first to successfully apply inference-time guidance to a masked model. This is challenging because Masked Transformers operates on their own learnable tokens, which differ from the codebook tokens utilized by the decoder. To address this, we propose Logits Editing for intermediate generation steps, allowing iterative adjustments to the generation process. In the final generation step, we directly update the embeddings in the codebook space through Codebook Editing. Further details on these approaches can be found in Section 3.3 (Inference-time Logits and Codebook Editing) and Supplementary Section A.10 (Dual-Space Categorical Straight-Through Estimator). This component also offers the additional benefit of being able to apply any arbitrary objective function, such as obstacle avoidance, which other methods (e.g., OmniControl, TLControl, MotionLCM) are not capable of.
>
>
> Note that all controllable motion models (e.g., GMD, OmniControl, TLControl) require some form of inference-time guidance to achieve accurate control. This is because motion data represents relative positions that depend on previous frame positions and rotations, whereas the control signal specifies in absolute positions. Unlike controllable models in the image domain, where pixel-to-pixel mapping is applicable, motion control faces these unique challenges. Further, working with masked tokens in latent space is more challenging compared to the motion space, as OmniControl. We also discuss these challenges in Section A.9: The Challenges of Motion Control Model.

---

> ### Author Response · Authors · 2024-11-22
> **______(2) Response to Reviewer KEsH______**
>
> ```
> Q2. Table 3 (L1 vs. L5) shows the control module performs not very well.
> ```
> **A2**:
> **Codebook Editing alone generates low quality (high FID)**, as shown in Table 3 (# 3): Codebook Editing directly perturbs the embedding based on the difference between the control signal's absolute position and the generated motion. This mechanism enables precise control over the trajectory by directly addressing discrepancies. However, because Codebook Editing modifies the motion after generation is complete, this post-hoc adjustment may impact the overall motion quality, especially if the generated motion deviates significantly from the control signal. **This demonstrates that Codebook Editing alone is insufficient for achieving high-quality results.**
>
> **Motion Control Model helps high quality generation (lower FID)**: The Motion Control Model improves motion quality by guiding the Masked Transformer generation process with the control signal at each layer. As a result, the generated tokens are already close to the desired outcome, requiring only small perturbation from Codebook Editing to achieve precise control with high quality.
>
> **Motion Control Model + Codebook Editing**: these two components complement each other—leading to improvements in both motion quality and trajectory control precision.
>
> From Table 3:
> | Method                              | FID ↓ (quality) | Trajectory Error ↓ |
> |-------------------------------------|-----------------|--------------------|
> | Codebook Editing (# 3)              | 0.190           | 0.0063             |
> | Motion Control Model (# 5)         | 0.128           | 0.3914             |
> | Codebook Editing + Motion Control Model (# 7) | 0.069  | 0.0005             |
>
>
>
> ```
> Q3. The authors miss two latest baselines, TLControl and MotionLCM.
> ```
> **A3**: The comparison between ControlMM and TLControl, is shown in Table 1, in terms of generation quality and control precision.  However, since TLControl is not open-sourced, we could not directly compare inference speeds in Figure 2. We also added MotionLCM to the main comparison Table 1 and Figure 2. Also, we visualize comparison in the [anonymous website](https://anonymous-ai-agent.github.io/CAM/).
> Since TLControl focuses on precise spatial control, this prioritization leads to lower motion quality, as reflected in its higher FID scores.
> MotionLCM emphasizes speed, enabling faster generation. However, this comes at the cost of even lower-quality motion (higher FID) and imprecise control.
> ControlMM overcomes these trade-offs by generating high-quality motion with precise control, while achieving real-time performance. The ControlMM-Fast variant generates 9.8 seconds of motion in just 4.94 seconds (in Table 5), without compromising quality. Moreover, the Accurate version of ControlMM achieves even more precise control while maintaining the same high-quality motion generation, setting it apart from existing methods.
>
> | Method            | Speed ↓  | FID ↓ (quality) | Trajectory Error ↓ | Loc. Err ↓ | Avg. Err ↓ |
> |-------------------|----------|-----------------|--------------------|------------|------------|
> | TLControl         | -        | 0.271           | 0.0000             | 0.0000     | 0.0108     |
> | MotionLCM         | 0.05 s   | 0.531           | 0.1887             | 0.0769     | 0.1897     |
> | ControlMM-Fast    | 4.94 s   | 0.059           | 0.0200             | 0.0075     | 0.0550     |
> | ControlMM-Accurate| 71.72 s  | 0.061           | 0.0000             | 0.0000     | 0.0098     |
>
>
> ```
> Q4 motions jitter
> ```
> **A4**: This is a common issue for rendering mesh which does not come from the model itself. Specifically, it arises during the retargeting process for the HumanML3D dataset. The issue is not unique to ours and affects the visualization of all models, including:
>
>
> - MotionLCM: latent diffusion
>    - https://dai-wenxun.github.io/MotionLCM-page/static/videos_mc_sparse/2.mp4  “a man crawls forward on his stomach”
>    - https://dai-wenxun.github.io/MotionLCM-page/static/videos_mc_sparse/3.mp4 “a person walks quickly and intentionally in zig-zag pattern forward.”
> - MDM: motion space diffusion
>    - https://guytevet.github.io/mdm-page/static/figures/kick1.mp4
>    - https://guytevet.github.io/mdm-page/static/figures/rope1.mp4
> - T2M-GPT: quantized token
>    - https://mael-zys.github.io/T2M-GPT/Ours/000066_pred.gif A man rises from the ground, walks in a circle ...
>    - https://mael-zys.github.io/T2M-GPT/Ours/004742_pred.gif a man starts off in an up right …
> ```
> Q5 generation quality
> ```
> **A5**: For quantitative analysis, the FID score measures generation quality, which is a key strength of our ControlMM model. Our model achieves FID score of 0.061, significantly lower than the 0.271, 0.218, and 0.531 achieved by OmniControl, TLControl, and MotionLCM, respectively. Additionally, we provide a qualitative comparison in Figure 5 and video demos comparison to SOTA on the website to further illustrate the advantages of our approach.

---

> ### Author Response · Authors · 2024-11-22
> **___(3) Response to Reviewer KEsH___**
>
> ```
> Q6 Whether the optimization in the latent space has some issues about the over-fitting or not is not well discussed.
> ```
> **A6**: Our model is trained and tested on HumanML3D, the motion covering  a broad range of human actions such as daily activities (e.g., 'walking', 'jumping'), sports (e.g., 'swimming', 'playing golf'), acrobatics (e.g., 'cartwheel') and artistry. From the evaluation on the test set, in Table 1, our model outperforming  other approaches on such a dataset indicates its superior generalization capabilities.
>
> In addition, all visualization demos on our project website are controlled by the synthetic spatial control signals that are not present in the training dataset or testing dataset. The high quality human motions based on the synthetic spatial control signals also showcase that our model is not overfit to the training dataset. Moreover, our model also enables Obstacle Avoidance and Body Part Timeline Control, tasks that the model has never encountered during the training phase.
>
>
> ___
> Dear Reviewer ```KEsH```, If our responses address your concerns, we kindly ask you to consider reevaluating this work and improving your rating. If there are remaining issues or additional guidance you can provide, we would be more than happy to address them.

---

> ### Author Response · Authors · 2024-11-24
> **______Additional Response to Reviewer KEsH______**
>
> Dear Reviewer KEsH,
>
> Thank you for your review. As the discussion deadline is approaching in two days, could you please check our response and let us know if anything remains unclear? If your concerns are resolved, we would appreciate it if you could consider reevaluating the work. Let us know if further clarification is needed.

---

> ### Author Response · Authors · 2024-11-26
> **______Additional Response to Reviewer KEsH -- visualization for "Component Analysis"______**
>
> We want to further clarify our response to your question ```Q2. Table 3 (L1 vs. L5) shows the control module performs not very well ... it seems that Codebook Editing is the main component for editing precision```
>
> We’ve added visualizations to the **Component Analysis** section of our anonymous demo website (https://anonymous-ai-agent.github.io/CAM/#component-analysis). These visualizations compare motions generated using only the **Codebook Embedding** versus those generated by the full model. When using only the Codebook Embedding, the motion quality is noticeably worse—it lacks realism and does not align well with the provided textual descriptions.
>
> Our **Motion Control Model** is specifically **designed to enhance motion quality**. As highlighted in **A2**, incorporating the Motion Control Model significantly improves motion quality, with the FID score improving from **0.190 to 0.069**.

---

> > ### Comment · Reviewer_KEsH · 2024-11-27
> > **My remaining concerns**
> >
> > Thanks for the response from the authors. I list my remaining questions as follows to finalize my rating.
> >
> > 1. For the "Motion Control Model", the authors claim that this is motivated by ControlNet. However, I notice that this has been claimed in MotionLCM. This difference is marginal for me currently. The comparison of the ControlNet part with that in MotionLCM seems not fair. To my understanding, the more reasonable choice is the L5 in Table 3. Right?
> >
> > 2. I still have concerns about the effectiveness of each part of the proposed methods. As pointed out in the review of  `d9b4`, the motion control module alone is not good enough, which motivated the development of the remaining modules.
> >
> > 3. As these components are not highly related to each other, the contribution of the proposed method is limited by this. Can you clarify this?
> >
> > 4. According to my questions in Q2, the comparison between L1-L5 is not mentioned in the response. I only see the comparison in L3-L5-L7.
> >
> > 5. As a researcher from the animation community, I am concerned about the speed of the algorithm. The controllable function is the main part that users might be concerned about during the interaction with the computer. If the algorithm is not efficient enough, I am curious how can this method be used by users. I value this is important because I notice MotionLCM treats this as an important motivation.
> >
> > 6. Considering the issue of overfitting, my concern is stated in my review. How are the hyper-parameters determined in your method? This is quite critical. Can the iteration be infinite? How can the step length be determined? I did not find this in the response.
> >
> > 7. I know the fitting might cause some jittering issues. However, I think the authors can provide the stickman motion visualization to resolve this concern. This process does not require the fitting. Otherwise, it is hard to evaluate whether the result is caused by the method.
> >
> > 8. I noticed that the revised Table 2 in the manuscript does not report the SOTA motion generation model MoMask. I would like to know the motivation. Besides, why not use the Momask as a baseline?
> >
> > I hope these remaining issues can be resolved, which are significant to determine the final rating. If any error, please point out directly.

---

> ### Author Response · Authors · 2024-11-30
> **______(1) Response to Reviewer KEsH Remaining Concerns______**
>
> Thank you reviewer ```KEsH``` for valuable feedback.
> ```
> Q1.1 I notice that ControlNet has been claimed in MotionLCM. This difference is marginal for me currently.
> ```
> **A1.1** We would like to clarify that in the related work section, we acknowledge OmniControl (Realism Guidance) and MotionLCM (Motion ControlNet) as the first to adopt a ControlNet-like structure. However, both OmniControl and MotionLCM are built on diffusion models, which is the framework that the original ControlNet was specifically designed for. In contrast, we are **the first to apply ControlNet-like to Generative Masked Model**. The detailed and unique implementation of applying ControlNet-like structure to Generative Masked Model is summarized as below.  Moreover, we are also **the first to introduce inference time guidance/optimization for the Generative Masked Model** which provides the additional advantage of enabling the application of arbitrary objective functions, such as obstacle avoidance—a task that both MotionLCM and OmniControl are not capable of. In addition, even through our work is designed for enabling ControlNet for Masked Motion Model.  **The techniques proposed in our work could be easily extended to generative masked image/video models**, such as MUSE, MAGVIT, and MEISSONIC.
>
> To further clarify the novelty and challenge of applying these two components to the Masked Model, we describe each component with additional details as follows:
>
> 1. **ControlNet-like (Motion Control Model)**: (1) **Underlying Working Principle**: Motion Control Model achieves spatial control by modifying the underlying motion token distribution. ControlNet realizes spatial control by alternating diffusion process. (2) **Network Architecture**: motion control model is a partial copy of the original masked motion model, where it does not possess the motion token classifier in the original model. ControlNet in OmniControl (Realism Guidance) and MotionLCM  (Motion ControlNet) has the identical architecture as the original diffusion model. Moreover, motion control model operates on the discrete motion tokens extracted from motion tokenizer. In contrast, OmniControl (Realism Guidance) operates in raw motion space and MotionLCM  (Motion ControlNet) operates in continuous latent space. (3) **Training Paradigm**: motion control model is trained using the combination of (i) generative masking training with consistency feedback, which has different training loss functions, compared with OmniControl  and MotionLCM  (ii) and training-time differential sampling, which are not required by diffusion models. (4) **Inference process**: because the generative masking training learned motion token distribution conditioned on text and spatial control prompts, during inference, motion control model follows the parallel token decoding that iteratively sample from the learned distribution to obtain the high-confident tokens, while remasking and repredicting low-confidence token in the following iterations. The challenge of applying ControlNet-like to Masked Model is that the model needs to decode [Mask] tokens for an initial motion token generation. However, [Mask] token only existing in Masked Transformer not the decoder from VQVAE. We overcome this by approximating the [Mask] token from codebook, as described in Sec A.9.
> 2. **Logits Editing + Codebook Editing**: We are the first to successfully apply inference-time guidance to a masked model. This is challenging because Masked Transformers operates on their own learnable tokens, which differ from the codebook tokens utilized by the decoder. To address this, we propose Logits Editing for intermediate generation steps, allowing iterative adjustments to the generation process. However, applying guidance directly to embeddings is impractical for Masked Models, as their Masked Transformers use learnable tokens that differ from the codebook space which requires for decoder of the Motion Tokenizer to reconstruct motion tokens to raw motion space as mentioned in Section A.10. In the final generation step, we directly update the embeddings in the codebook space through Codebook Editing. These components also offer the additional benefit of being able to apply any arbitrary objective function, such as obstacle avoidance, which other methods (e.g., OmniControl, TLControl, MotionLCM) are not capable of.

---

> ### Author Response · Authors · 2024-11-30
> **______(2) Response to Reviewer KEsH Remaining Concerns______**
>
> ```
> Q1.2 More reasonable choice is to compare ControlNet-like of ControlMM (L5 in Table 3) to MotionLCM
> ```
> **A1.2** We respectfully disagree. Limiting the comparison to only certain component to MotionLCM may be not very reasonable. Current SOTA methods commonly utilize ControlNet-like modules and/or inference-time guidance. However, by design, MotionLCM is fundamentally different as it prioritizes speed by distilling the diffusion process into just 1–4 steps. This contrasts with methods like GMD and OmniControl, which require 1,000 steps for guided diffusion. Utilizing only 1-4 step guided diffusion is impractical for MotionLCM. Additionally, TLControl does not employ a ControlNet-like architecture either. Therefore, restricting comparisons solely to architectures with ControlNet-like components would make direct comparisons across baseline methods infeasible.
>
> For clarity, the breakdown of components, along with FID (quality) and trajectory error (precision control), is provided in the table below.
>
> | | Name of ControlNet-like Component|Name of Inference Time Guidance Component| Architecture| FID ↓ | Trajectory Error ↓ |
> |-|-|-|-|-|-|
> |GMD (ICCV 2023)| N/A|Guided Motion Diffusion|Diffusion|0.576|0.0931|
> |Omnicontrol (ICLR 2024)| Realism Guidance| Spatial Guidance| Diffusion| 0.218| 0.0387|
> |MotionLCM (ECCV 2024)| Motion ControlNet|N/A|Diffusion|0.531|0.1887|
> |TLControl (ECCV 2024)| N/A| Runtime Optimization|Single step feed forward|0.271| 0.0000|
> |**ControlMM**|Motion Control Model|Logits + Codebook Editing|**Masked Model**|**0.061**|**0.0000**|
>
> ```
> Q.2 Effectiveness of Motion Control Model, As pointed out by reviewer d9b4 Motion Control Model alone is not good enough
> ```
> **A2**  We addressed this concern for reviewer ```d9b4```, who is satisfied with our response. We hope this clarification also addresses your concern. In sum, Motion Control Model and Codebook/Logits Editing are complementary components and combining them leads to the optimal motion quality and trajectory control precision. Motion Control Model is training-time optimization that improve motion quality/fidelity by learning joint categorical distribution of motion tokens, while guiding the generated motion to closely align with the input spatial control signals and text prompts.  As a result, the generated motion is required only small perturbation from Codebook/Logits Editing during inference to realize more precise control.
>
> Reviewer ```d9b4``` concerned that "OmniControl does not seem to use any inference-time optimization (only ControlNet-like part)". However, we addressed that OmniControl also utilizes comparable architectures **ControlNet-like (Realism Guidance in OmniControl and Motion Control Model in ControlMM)** and **Inference Time Guidance (Spatial Guidance in OmniControl and Logits + Codebook Editing in ControlMM)**. Moreover, the table below compares only the ControlNet-like modules of OmniControl (Realism Guidance) and ControlMM (Motion Control Model). Our Motion Control Model demonstrates superior performance, showing better quality (FID and R-Precision) and more precise control (Trajectory, Location, and Average Error).
>
> |Method|R-Precision Top-3 ↑| FID ↓|Diversity →| Foot Skating Ratio ↓|Trajectory Error ↓|Location Error ↓|Average Error ↓|
> |--|--|--|--|--|--|--|--|
> | Omnicontrol (only Realism Guidance)|0.691| 0.351|**9.506**|**0.0561**|0.4285| 0.2572| 0.4137|
> | ControlMM (only Motion Control Model)|**0.802**|**0.128**|9.475|0.0594|**0.3914**|**0.2400**|**0.4041**|

---

> ### Author Response · Authors · 2024-11-30
> **______(3) Response to Reviewer KEsH Remaining Concerns______**
>
> ```
> Q3. Components are not highly related to each other, the contribution of the proposed method is limited by this.
> ```
> **A3.** Our model consists of Motion Control Model and Logits+Codebook Editing, each with its own benefit. Combining these components is not a limitation, but rather a complementary approach that helps improve both quality  and precision control, as shown in the table below. We explain each component as follows:
>
> 1. **ControlNet-like (Motion Control Model)** The purpose of this module is to ensure high-quality motion generation by connecting the control signal module to each layer of the Masked Motion Model. This allows the model to predict probability distributions of the codebook that are close to the control signal. While Logits + Codebook Editing can perturb the logits far away from the true distribution, the Motion Control Model helps guide the model to output logits that are close to the true distribution, as it is trained to output logits w.r.t. to the true distribution of ground truth tokens obtained from VQVAE.
> 2. **Inference Time Guidance (Logits + Codebook Editing):**  Logits Editing and Codebook Editing can be considered the same component, as they utilize the exact same optimization function. The optimize function updates logits during earlier unmasking steps (Logits Editing), while the codebook embeddings is updated during the final unmasking step (Codebook Editing). Optimizing logits in the early stage can help to maintain high quality while optimizing codebook embedding in the last stage helps improve precise control. Moreover, because it can incorporate any optimization function during inference, this component is versatile and can adapt to arbitrary tasks such as obstacle avoidance.
>
> The advantage of Logits Editing and the Motion Control Model is their ability to maintain motion quality (R-Precision, FID, and Foot Skating), while the advantage of Codebook Editing lies in its ability to achieve precise control (Trajectory Error, Location Error, and Average Error). By combining these components, ControlMM achieves state-of-the-art performance in both quality and precision control.
> From Table 3:
> |||Quality|             |                   |       Control Error           |                 |
> |-------------------------------------|----------------------------------|------------------------|--------------------------|-------------------|------------------|-----------------|
> |              Components                        | R-Precision Top-3 ↑              | FID ↓                  | Foot Skating Ratio ↓     | Trajectory Error ↓ | Location Error ↓ | Average Error ↓ |
> | Motion Control Model + Logits Editing (#6) | **0.814**                         | **0.051**                  | **0.0541**                   | 0.1302            | 0.0623           | 0.1660          |
> | Codebook Editing (#3)      | 0.786                         | 0.190                  | 0.0616                   | 0.0063            | 0.0005           | 0.0283          |
> | ControlMM (Full Model)              | 0.809                         | 0.061                  | 0.0547                   | **0.0000**            | **0.0000**          | **0.0098**          |
>
>
>
>
>
>
> ```
> Q4 comparison between L1-L5
> ```
> **A4.** We assume that reviewer ```KEsH```'s question is about the effectiveness of the Motion Control Model (L5). We have addressed this in ```Q2``` by comparing it to OmniControl. Moreover, a comparison between the baseline control (L1) with the Motion Control Model (L5) may not provide a clear picture. Instead, we answer by comparing the full model with and without the Motion Control Model.
> The purpose of the Motion Control Model is to maintain the quality of generation. As shown in the table below, without the Motion Control Model, the FID score worsens significantly from 0.061 to 0.142.
> | Components| R-Precision Top-3 ↑ | FID ↓ (quality) | Foot Skating Ratio ↓ | Trajectory Error ↓ | Location Error ↓ | Average Error ↓ |
> |--------|---------------------|-----------------|-----------------------|---------------------|-------------------|-----------------|
> | Without Motion Control Model (#4)   | 0.795| 0.142| 0.0577| 0.0032| 0.0002| 0.0218|
> | ControlMM (Full Model)| 0.809| 0.061| 0.0547| 0.0000| 0.0000| 0.0098|

---

> ### Author Response · Authors · 2024-11-30
> **______(4) Response to Reviewer KEsH Remaining Concerns______**
>
> ```
> Q5. Speed of the algorithm. How can this method be used by users. MotionLCM treats this as an important motivation.
> ```
>
> **ControlMM-Fast  is ~20x faster and more accurate compared to other SOTA**
>
> ControlMM-Accurate is relatively faster than SOTA methods (GMD and OmniControl) while delivering significantly better precision control (measured by Trajectory Error, Location Error, and Average Error) and higher quality (measured by R-Precision and FID).
> - ControlMM-Fast achieves even greater speed, completing motion generation in 4.94 seconds, compared to 87.33 seconds for OmniControl and 132.49 seconds for GMD. With this speed ControlMM already enables real-time generation, producing 9.8 seconds of motion in just 4.94 seconds. All models are tested on the same NVIDIA A100 GPUs.
> - With such high generation speed, ControlMM-fast also maintains better quality, with an FID of 0.059, compared to 0.218 (OmniControl) and 0.576 (GMD). It also outperforms in precision control, achieving a 2% Trajectory Error, compared to 3.87% (OmniControl) and 9.31% (GMD).
> Note that the number of iterations in Logits + Codebook Editing can be adjusted during inference to trade off between speed and precision control without affecting motion quality (FID and R-Precision). As the table shown below, ControlMM-Accurate can achieve Average Error of 0.98 cm and 0.00% of Trajectory Error, while still faster than OmniControl and GMD.
>
>
> **MotionLCM focuses on speed, which comes at the cost of lower quality and precision control.**
>
> - **Motion Quality**:  our ControlMM  achieves a FID score of 0.059 (ControlMM-fast), outperforming MotionLCM which has a FID score of 0.531.
>
> - **​​Control Precision**: MotionLCM  has an average joint error of 18.97 cm, while our ControlMM has an average joint error of 0.0098. Additionally, according to the Trajectory Error metric, MotionLCM exhibits joint errors exceeding 50 cm in 18.87% of its generated motions, compared to 2.00% and 0.00% for our Fast and Accurate versions, respectively.
> As shown in demos on the https://anonymous-ai-agent.github.io/CAM/, ControlMM also shows higher quality and more precise control in generation motion, compared with MotionLCM. For more visualization compared to MotionLCM, we also uploaded the visualization comparing MotionLCM to our ControlMM in the "Supplementary Material" zip file, located in the 'ControlMMvsMotionLCM' folder.
>
> **Real-World Application**
> ControlMM is designed to generate high-quality, precise motion generation with real-time performance, making it practical for real-world applications.  MotionLCM prioritizes speed at the expense of precision and quality.
>
> | Method                  | Speed ↓   | R-Precision Top-3 ↑ | FID ↓ | Trajectory Error [>50 cm] (%) ↓ | Loc. Err [>50 cm] (%) ↓ | Avg. Err. (cm) ↓ |
> |-------------------------|-----------|---------------------|-----------------|---------------------------------|--------------------------|------------------|
> | GMD                     | 132.49 s  | 0.665               | 0.576           | 9.31%                           | 3.21%                    | 14.39 cm         |
> | OmniControl             | 87.33 s   | 0.687               | 0.218           | 3.87%                           | 0.96%                    | 3.38 cm          |
> | MotionLCM               | 0.05 s    | 0.752               | 0.531           | 18.87%                          | 7.69%                    | 18.97 cm         |
> | ControlMM-Fast          | 4.94 s    | 0.808               | 0.059           | 2.00%                           | 0.75%                    | 5.50 cm          |
> | ControlMM-Accurate      | 71.72 s   | 0.809               | 0.061           | 0.00%                           | 0.00%                    | 0.98 cm          |
>
>
> For more details on ControlMM-Fast and ControlMM-Accurate, refer to Table 5 in the manuscript.

---

> ### Author Response · Authors · 2024-11-30
> **______(5) Response to Reviewer KEsH Remaining Concerns______**
>
> ```
> Q6.  hyper-parameters, Can the iteration be infinite?
> ```
> **A6.** We detail the hyperparameters in the manuscript (Sec. A3). The default settings use 100 iterations for Logits Editing and 600 iterations for Codebook Editing with learning rate 6e-2. We conducted a parameter search on the evaluation set to determine the minimum number of iterations required to reduce the Location Error to zero. With the limit of iteration, the optimization process does not run indefinitely. However, these iterations can be fewer if the loss reaches zero before the maximum iteration limit.
> Moreover, the table below demonstrates that increasing the number of iterations for Codebook Editing continues to improve the Average Error without any signs of degradation in generation quality (FID).
> | # of Iteration of Codebook Editing | R-Precision Top-3 ↑ | FID ↓ | Diversity → | Foot Skating Ratio ↓ | Traj. Err. (50 cm) ↓ | Loc. Err. (50 cm) ↓ | Avg. Err. ↓ |
> |-------------------------------------|---------------------|-------|-------------|-----------------------|----------------------|---------------------|-------------|
> | 600                                 | 0.809               | 0.061 | 9.496       | 0.0547                | 0.0000               | 0.0000              | 0.0098      |
> | 1200                                | 0.809               | 0.061 | 9.517       | 0.0551                | 0.0000               | 0.0000              | 0.0063      |
>
>
> ```
> Q7. Stickman motion for motion jittering
> ```
> Thank you for your suggestion. We have also uploaded the skeleton motion in the "Supplementary Material" zip file under the 'jittering' folder. The skeleton visualization shows no jittering issue, which confirms that the issue arises during the retargeting process for the HumanML3D dataset and not from our model.
>
> ```
> Q8. revised Table 2 does not report MoMask
> ```
> Note that Table 2 has not been revised and remains the same as the first version. MoMask cannot control joint positions because it encodes and quantizes all body joints into a single token. As a result, modifying only the upper body is not possible with MoMask. In contrast, MMM (Masked Motion Model) addresses this limitation by training separate codebooks for the upper and lower body, allowing more control over upper body.

---

> ### Author Response · Authors · 2024-12-03
> **______Additional Response to Reviewer KEsH______**
>
> Dear Reviewer KEsH,
>
> Thank you for your review. Could you please check our response and let us know if anything remains unclear? If your concerns are resolved, we would appreciate it if you could consider reevaluating our work.
>
> > Note that we already added ***MotionLCM*** to Table1, Figure 2, Figure 6, and demo website.

---

> ### Author Response · Authors · 2024-12-04
> **______SUMMARY OF ANSWER TO REVIEWER KEsH______**
>
> Since we did not receive final response from reviewer ```KEsH```, we summarize the "My remaining concerns" of reviewer ```KEsH``` as follows:
>
> Note that the questions from reviewer ```KEsH``` only compare with **MotionLCM**, which is a 1-step diffusion method without guided diffusion, making it not directly comparable to our ControlMM. It is also helpful to see our answers to similar questions raised by other reviewers (summarized in the **"Similar Questions Summary"** at the very top post), as this should provide a clearer picture.
> ```
> Q1.1 ControlNet-like has been claimed in MotionLCM
> ```
> - Our ControlMM is the first to introduce [1] Motion Control Module (ControlNet-like) and [2] Logits/Codebook Editing (Inference Time Guidance) to the Masked Motion Model. In contrast, MotionLCM is similar to OmniControl which both utilize ControlNet-like on diffusion-based models.
> ```
> Q1.2 Should only compare Motion Control Module (ControlNet-like) to MotionLCM.
> Q2 Motion Control Module (ControlNet-like) alone does not perform well (​​compared to MotionLCM)
> Q3 Motion Control Module (ControlNet-like) and ​​Logits/Codebook Editing (Inference Time Guidance) are not highly related to each other
> Q4 Comparison of Motion Control Module (ControlNet-like) alone
> ```
> - All SOTA methods, i.e., GMD, OmniControl, and TLControl, utilize some form of Inference Time Guidance to achieve high-precision control. Moreover, GMD and TLControl do not use a ControlNet-like architecture. If the comparison is restricted to only ControlNet-like components, none of the baseline methods can be directly compared.
> - Motion Control Module is designed for maintaining quality, combined with Logits/Codebook Editing (Inference Time Guidance) which is designed for high precision control which leads to the SOTA results in both quality and precision control. Additionally, Motion Control Module (ControlNet-like) alone performs better than OmniControl with ControlNet-like part.
> - Please refer to the table in answer _“A1.2”_ which includes a comparison of all components of SOTA methods to provide a complete overview.
> ```
> Q5 MotionLCM is faster so is more practical
> ```
> - Our ControlMM achieves SOTA performance across all metrics, while maintaining real-time performance, and is still faster than other methods.
> - MotionLCM focuses only on speed, which causes it to perform the worst in all other metrics.
> - Please see our demo https://anonymous-ai-agent.github.io/CAM and "Supplementary Material" zip file, located in the 'ControlMMvsMotionLCM' folder. MotionLCM quality and control precision are significantly worse which make ControlMM more practical.
> ```
> Q6. Can the iteration be infinite?
> ```
> - We use a fixed number of iterations so it will not run infinitely.
> ```
> Q7. jittering issues
> ```
> - The skeleton motion visualization in the "Supplementary Material" zip file, located in the 'jittering' folder, shows no jittering issues, confirming that the problem does not originate from our model.
> ```
> Q8 Not report MoMask
> ```
> - MoMask does not have controllability.

---

### Official Review · Reviewer_gWgL · 2024-10-26

**Soundness:** 3
**Presentation:** 3
**Contribution:** 3
**Rating:** 6
**Confidence:** 4

**Summary:**

This paper presents ControlMM, a framework for integrating spatial control signals into a masked motion model. The main innovations focus on two aspects: (1) Masked Consistency Modeling to ensure high-fidelity motion by minimizing inconsistencies between input control signals and generated motion, and (2) Inference-time Logit Editing to enhance control precision and facilitate tasks like obstacle avoidance without retraining. The experimental results show that ControlMM delivers strong performance in motion quality, control accuracy, and generation speed, while supporting diverse applications, such as arbitrary joint and frame control and body part timeline control.

**Strengths:**

1. This paper extends the ControlNet concept from a diffusion-based framework to a mask generation framework. In addition, the Inference-time Logit Editing feature is presented as an innovative contribution.

2. The paper presents the methodology in a well-organized and accessible way, making both the theoretical framework and implementation details easy to understand.

3. The paper provides extensive experiments to validate the performance of ControlMM. These include comparisons with other models and ablation studies that highlight the contributions of specific components (e.g., masked consistency modeling, logit editing).

**Weaknesses:**

1. While the paper conducts extensive experimentation on large datasets like HumanML3D, it does not explore how the model performs under data-constrained scenarios. Many other studies in the field validate their models on smaller datasets, such as KIT, which provide insights into the robustness of models in resource-limited settings.

2. Although the paper compares ControlMM with several state-of-the-art models, such as OmniControl and GMD, it lacks comparisons with MoMask[1]—an earlier exploration of the motion mask generation framework. Including this comparison in Table 1 is essential, as MoMask forms the baseline upon which ControlMM’s controllability innovations build. Clarifying the specific contribution of ControlMM to the improvement of the masked generation framework would provide a more complete evaluation.

3. While the paper provides strong quantitative results, it lacks qualitative user studies similar to those conducted in other works, such as MLD[2].

[1] Chuan Guo, Yuxuan Mu, Muhammad Gohar Javed, Sen Wang, and Li Cheng. Momask: Generative masked modeling of 3d human motions. In Proceedings of the IEEE/CVF Conference on Computer Vision and Pattern Recognition (CVPR), 2024.

[2] Chen Xin, Biao Jiang, Wen Liu, Zilong Huang, Bin Fu, Tao Chen, Jingyi Yu, and Gang Yu. Executing your commands via motion diffusion in latent space. In Proceedings of the IEEE/CVF Conference on Computer Vision and Pattern Recognition (CVPR), 2023.

**Questions:**

1. We noticed that the paper relies heavily on specific joint control during both training and testing. Could you clarify whether this joint control data is also used during test set validation? If so, would this constitute a data leakage issue? Additionally, how would the system determine which joint information to use for control in practical applications, where such data may not be known in advance?

2. As a generative model, ControlMM’s performance should not only be assessed on quality and controllability but also on diversity. Can the authors provide additional qualitative or quantitative examples that showcase the diversity of generated motions under the same control constraints (e.g., identical control signals and prompts)? This would help demonstrate that the model can generate varied outputs without overfitting to specific patterns.

---

> ### Author Response · Authors · 2024-11-22
> **______(1) Response to Reviewer gWgL______**
>
> Thank you for your time and valuable feedback. Below, we address your questions and comments:
> ```
> Q1. comparisons with MoMask
> ```
> **A1**. MoMask itself lacks the ability to control joint positions, which makes it not directly comparable in Table 1. Note that the original MDM also does not provide controllability. However, OmniControl reports results for MDM by applying a simple imputation strategy: it directly fills in the control joints information and allows MDM to diffuse the motion following control joins. This approach works for MDM because it operates directly in the motion space, where such imputation is straightforward. In contrast, this method is not applicable for quantized tokens in latent like MoMask.
>
> ```
> Q2. Could you clarify whether this joint control data is also used during test set validation? If so, would this constitute a data leakage issue?
> ```
> **A2**. Yes, we use the ground truth joint information during both training and testing, following the settings of GMD and OmniControl. However, this is not data leakage. Simply replacing control joints with the motion directly will result in poor motion quality (high FID) since the motion will not be consistent with generated joints. Our experiments show that our ControlMM can balance between both precision control and high quality.
> Note that the dense signal control demonstrated on the website does not come from ground truth data in the dataset. These signals are synthesized, such as controlling the right hand to draw a heart in “the person draws a heart with hand” or defining a trajectory using a sinusoidal wave forming a circle in “a person walks in a circle clockwise.” These examples illustrate the model's ability to handle arbitrary control signals beyond the training data.
> ```
> Q3. How would the system determine which joint information to use for control in practical applications, where such data may not be known in advance?
> ```
> **A3**. The trajectory control can be synthesized to demonstrate various patterns, such as walking in a circular or zigzag manner, or guiding hand movements to draw specific shapes. Additionally, it can be adapted for sparse signal control, such as in a goal-reaching task where only the final frame is constrained by the control signal. These examples highlight the model's flexibility in handling both dense and sparse control signals.
> ```
> Q4. KIT Dataset
> ```
> **A4** We also tested ControlMM on the KIT dataset and compared it to state-of-the-art (SOTA) methods. Despite the KIT dataset being significantly smaller than HumanML3D, ControlMM consistently outperformed other SOTA methods in both quality and precise control, demonstrating its robustness in resource-limited settings. We added KIT dataset result in Section "A.12 KIT DATASET".
> | **Method**| **R-Precision Top-3 ↑** | **FID ↓** | **Diversity →** | **Traj. Err. (50 cm) ↓** | **Loc. Err. (50 cm) ↓** | **Avg. Err. ↓** |
> |-|-|-|-|-|-|-|
> | **PriorMDM**| 0.397| 0.851| 10.518| 0.3310| 0.1400|0.2305|
> | **GMD**| 0.382| 1.565| 9.664| 0.5443| 0.3003| 0.4070|
> | **OmiControl**| 0.397| 0.702| **10.927**| 0.1105| 0.0337| 0.0759|
> | **TLControl** | **0.757**| 0.432| 10.723| 0.0028| 0.0011| 0.0276|
> | **ControlMM** | 0.747| **0.378** | 10.527| **0.0018**| **0.0001**| **0.0160**|
> ```
> Q5 Qualitative examples that showcase the diversity
> ```
> **A5** We added diversity generation under the same prompt for both signal control joints and object avoidance. These visualizations have been added to the **Diversity** section of our anonymous website demo https://anonymous-ai-agent.github.io/CAM/.
> ```
> Q5. User study
> ```
> **A5** We conducted pairwise comparisons in the user study by asking participants to answer two questions: “Which motion is more realistic (better quality)?” and “Which motion is generated with more precise control?” The motions were generated from 30 randomly selected text descriptions from the test set of the HumanML3D dataset. We engaged 6 participants and presented them with three pairwise comparisons: our method versus OmniControl, our method versus GMD, and our method versus MotionLCM. Our ControlMM was rated significantly more realistic and demonstrated better precision control compared to other state-of-the-art methods.
> | Comparison| Realistic | Precision Control |
> |-|-|-|
> | ControlMM vs OmniControl | 78%| 72%|
> | ControlMM vs GMD| 82%| 79%|
> | ControlMM vs MotionLCM| 88%| 94%|

---

> ### Author Response · Authors · 2024-11-30
> **______Additional Response to Reviewer gWgL______**
>
> Dear Reviewer gWgL,
>
> Thank you for your review. As the discussion deadline is approaching in two days, could you please check our response and let us know if anything remains unclear? If your concerns are resolved, we would appreciate it if you could consider reevaluating the work. Let us know if further clarification is needed.

---

### Official Review · Reviewer_d9b4 · 2024-10-31

**Soundness:** 3
**Presentation:** 3
**Contribution:** 2
**Rating:** 6
**Confidence:** 4

**Summary:**

This paper presents an approach, namely ControlMM, for controllable motion generation. ControlMM is based on a generative masked motion model. A ControlNet-like controlling module is introduced to inject spatial guidance. Additionally, inference-time logits and embedding optimization are proposed to reinforce the spatial control signal as post-processing. Extensive experiments are performed to show the sota performance on motion generation and control ability.

**Strengths:**

+ The ControlNet-like design is first used in the masked generation model, which is interesting and inspiring.
+ The inference-time optimization has reasonable performance on control joint error. More impressively, the generation quality during optimization does not drop.
+ The overall performance is good compared with baseline methods.

**Weaknesses:**

- My biggest concern is that from the ablation study, the motion control module along does not seem to perform well. In fact, from table 3, the fifth variation has the second worst trajectory error. Only using inference-time optimization on a vanilla motion generation model achieves better trajectory error. This undermines the contribution of the controlnet-like motion control module.
- Correct me if I am wrong, but OmniControl does not seem to use any inference-time optimization for better control accuracy. Therefore, the comparison seems not fair.
- Minor Writing Issues:
1. Line93: i.e. => \textit{i.e.},
2. Line94: Control => control
3. Line101: use \citep

**Questions:**

See my comments in the weaknesses section. If my concern towards the effectiveness of the motion control module can be addressed, I would consider raise the rating.

---

> ### Author Response · Authors · 2024-11-22
> **______Response to Reviewer d9b4______**
>
> Thank you for your time and valuable feedback. Below, we address your questions and comments:
> ```
> Q1. Motion Control Module alone does not seem to perform well.
> ```
> **A1**: **Codebook Editing alone generates low quality (high FID)**: Codebook Editing directly perturbs the embedding based on the difference between the control signal's absolute position and the generated motion. This mechanism enables precise control over the trajectory by directly addressing discrepancies. However, because Codebook Editing modifies the motion after generation is complete, this post-hoc adjustment may impact the overall motion quality, especially if the generated motion deviates significantly from the control signal.
>
> **Motion Control Model helps high quality generation (lower FID)**: The Motion Control Model improves motion quality by guiding the Masked Transformer generation process with the control signal at each layer. As a result, the generated tokens are already close to the desired outcome, requiring only small perturbation from Codebook Editing to achieve precise control with high quality.
>
> **Motion Control Model + Codebook Editing**: these two components complement each other—leading to improvements in both motion quality and trajectory control precision.
>
> From Table 3:
> | Method                              | FID ↓ (quality) | Trajectory Error ↓ |
> |-------------------------------------|-----------------|--------------------|
> | Codebook Editing (# 3)              | 0.190           | 0.0063             |
> | Motion Control Model (# 5)         | 0.128           | 0.3914             |
> | Codebook Editing + Motion Control Model (# 7) | 0.069  | 0.0005             |
>
> ```
> Q2: OmniControl does not seem to use any inference-time optimization
> ```
> **A2: Spatial Guidance** in OmniControl is guided diffusion, which is an inference-time optimization approach that optimizes the diffusion noise. Different from OmniControl，our method relies on the masked motion model that does not leverage the diffusion process. Therefore, we cannot apply the noise optimization scheme adopted by OmniControl. Instead, our method optimizes the logits (i.e., unnormalized probabilities of motion tokens) and codebook embeddings to enhance spatial controllability. Therefore, I believe the comparison with OmniControl is fair.
> Even Though Spatial Guidance in OmniControl updates noise only once in each diffusion step, it requires 1000 diffusion steps. Comparing to our “Fast” version which requires only 100 iteration for Codebook Editing already achieves better Trajectory Error while significantly faster and higher quality (lower FID)
> From Table 5:
> | Method          | Speed ↓ | FID ↓ (quality) | Trajectory Error ↓ |
> |-----------------|---------|-----------------|--------------------|
> | ControlMM-Fast  | 4.94 s  | 0.0590          | 0.0200             |
> | OmniControl     | 87.33 s | 0.218           | 0.0387             |
>
> Note that all controllable motion models (e.g., GMD, OmniControl, TLControl) require some form of inference-time guidance to achieve accurate control. This is because motion data represents relative positions that depend on previous frame positions and rotations, whereas the control signal specifies in absolute positions. Unlike controllable models in the image domain, where pixel-to-pixel mapping is applicable, motion control faces these unique challenges. Furthermore, working with masked tokens in quantized latent space as ControlMM is more challenging compared to the motion space as OmniControl. We also discuss these challenges in Section A.9: The Challenges of Motion Control Model.
> ___
> Dear Reviewer ```d9b4```, If our responses address your concerns, we kindly ask you to consider reevaluating this work and improving your rating. If there are remaining issues or additional guidance you can provide, we would be more than happy to address them.

---

> ### Author Response · Authors · 2024-11-24
> **______Additional Response to Reviewer d9b4______**
>
> Dear Reviewer d9b4,
>
> Thank you for your review. As the discussion deadline is approaching in two days, could you please check our response and let us know if anything remains unclear? If your concerns are resolved, we would appreciate it if you could consider reevaluating the work. Let us know if further clarification is needed.

---

> > ### Comment · Reviewer_d9b4 · 2024-11-26
> > **Thanks for the rebuttal. The rating has been updated.**
> >
> > I thank authors for the detailed rebuttal. My concerns have been addressed. Therefore, I have updated my rating to acceptance.

---

### Official Review · Reviewer_9WzV · 2024-11-03

**Soundness:** 2
**Presentation:** 4
**Contribution:** 4
**Rating:** 6
**Confidence:** 4

**Summary:**

Text-driven human motion generation has received attention but faces challenges in precise spatial control. Existing controllable motion generation models have difficulties in generating high-fidelity motion with real-time inference and handling both sparse and dense spatial control signals. This paper presents an interesting and innovative approach to controllable text-to-motion generation. The proposed ControlMM method shows promising results in terms of motion quality, control precision, and generation speed, outperforming existing state-of-the-art methods in many aspects.

**Strengths:**

1. This paper achieves real-time, high-fidelity, and high-precision controllable motion generation. Besides, it outperforms state-of-the-art methods in motion quality (better FID scores) and control precision (lower average error).
2. This paper ntroduces masked consistency modeling and inference-time logit editing. Masked consistency modeling ensures high-fidelity motion generation and reduces inconsistency between input and extracted control signals. Inference-time logit editing allows for better control precision and enables new control tasks.
3. This framework generates motions 20 times faster than diffusion-based methods.
4. Based on this design choice, it enables a wide range of applications such as any joint any frame control, body part timeline control, and obstacle avoidance.

**Weaknesses:**

1. The model's architecture and training process appear to be relatively complex. The use of multiple components and techniques may make it difficult for other researchers to understand and implement the model. A more detailed and intuitive explanation of the model's inner workings could be beneficial for reviewers.

2. While the paper mentions some handling of out-of-distribution situations in the context of body part timeline control, it is not clear how well the model would perform in more extreme or unforeseen out-of-distribution cases. This could be a potential limitation in real-world applications where the input data may deviate significantly from the training data.

3. The reliance on pretrained models (e.g., MoMask) may limit the model's adaptability and performance in some cases. If the pretrained models have biases or limitations, these could be carried over to the ControlMM model. It would be interesting to see if the model can be trained from scratch or with different pretrained models and still achieve similar performance.

**Questions:**

1. Provide a more detailed and intuitive explanation of the model's architecture and training process. This could include visualizations or step-by-step breakdowns of how the different components interact to generate motions.
2. Conduct more experiments to test the model's robustness in handling a wider range of out-of-distribution data. This could involve creating synthetic datasets with more extreme variations or using real-world datasets from different domains to evaluate the model's performance.

---

> ### Author Response · Authors · 2024-11-22
> **______Response to Reviewer 9WzV______**
>
> Thank you for your time and valuable feedback. Below, we address your questions and comments:
> ```
> Q1: A more detailed and intuitive explanation of the model's inner workings could be beneficial for reviewers.
> ```
> **A1**:  We updated figure 3 for better visualization of the motion control model, the key element of our approach. In addition, we added more implementation details in section 3 and supplementary materials. Intuitively speaking ,ControlMM is based on generative masked motion model, which is very similar to BERT-like language model that leverages token masking and reconstruction to learn the probabilistic distribution of human motion sequence. The existing generative masked motion models do not have spatial controllability, while exhibiting high-quality motion generation capacity. This paper aims to integrate spatial control into the generative masked motion model. Our approach has three key components.  **Motion Control Model** is a training time optimization approach that aims to learn the accurate mapping from the text prompt and the spatial control prompt to the motion sequence by fine-tuning the pretrained generative masked motion model via a ControlNet-inspired approach. **Logits Editing + Codebook Editing** These are analogous to Guided Diffusion, which applies inference-time guidance. However, adapting this concept to Masked Models introduces a unique challenge: the embeddings of the Masked Transformer and the decoder from VQVAE operate in different spaces. To address this, we update the logits (not embeddings) during the initial unmasking steps—a process we call Logits Editing. At the final step, we directly update the embeddings in the codebook space, referred as “Codebook Editing”
>
> ```
> Q2: How well the model would perform in more extreme or unforeseen out-of-distribution cases
> ```
> **A2**: Our demo on the website demonstrates that ControlMM effectively handles out-of-distribution (OOD) scenarios across various tasks. Note that all dense signal controls presented in the demo are synthesized, and are not in the dataset. For instance, it can generate creative motions such as "a person draws a heart with their hand" with heart-shape trajectory or "a person walks in a circle clockwise" following syntactic trajectories like a sinusoidal circle wave. These examples illustrate the model's ability to handle arbitrary control signals beyond the training data. Additionally, the model successfully tackles tasks like Body Part Timeline Control and Obstacle Avoidance, even though it was never explicitly trained on these tasks, demonstrating its ability to adapt to arbitrary objective functions.
>
> ```
> Q3: It would be interesting to see if the model can be trained from scratch or with different pretrained models and still achieve similar performance.
> ```
> **A3**: This is interesting question. Note that MoMask is  the current state-of-the-art Masked Motion Model. Experimenting with other models may result in a performance drop. So we investigate three different training strategies: (1) Frozen MoMask: MoMask base model is frozen  and motion control model (MCM) is trained with the initial weight copied from the base model, (2)  Frozen MoMask + MCM scratch: MoMask base model is  frozen, and motion control model is trained from scratch. (3 ) MoMask scratch + MCM scratch:  MoMaks and Motion Control Model are jointly trained from  scratch. As shown in  the table below, jointly  training MoMask and MCM from scratch leads to significantly degraded quality (i.e., much higher FID). By keeping the MoMask base model frozen, training the MCM from scratch or from the initial weights copied from yields the best and similar momotion generation quality (i.e., much lower FID)
> | Method                              | R-Prec. Top-3 ↑ | FID ↓                  | Diversity → | Foot Skat. ↓ | Trajectory Error ↓ | Loc. Err ↓ | Avg. Err. ↓ |
> |-------------------------------------|---------------------|-----------------|-------------|----------------------|---------------------|------------|-------------|
> | Frozen MoMask + MCM copy weights    | 0.809               | 0.061           | 9.496       | 0.0547               | 0.0000              | 0.0000     | 0.0098      |
> | MoMask scratch + MCM scratch        | 0.720               | 0.780           | 9.30714     | 0.05177              | 0.0001              | 0.0000     | 0.0087      |
> | Frozen MoMask + MCM scratch         | 0.810               | 0.062           | 9.50707     | 0.0545               | 0.0000              | 0.0000     | 0.0099      |
>
> ___
> ___
> Dear Reviewer ```9WzV```, If our responses address your concerns, we kindly ask you to consider reevaluating this work and improving your rating. If there are remaining issues or additional guidance you can provide, we would be more than happy to address them.

---

> ### Author Response · Authors · 2024-11-24
> **______Additional Response to Reviewer 9WzV______**
>
> Dear Reviewer 9WzV,
>
> Thank you for your review. As the discussion deadline is approaching in two days, could you please check our response and let us know if anything remains unclear? If your concerns are resolved, we would appreciate it if you could consider reevaluating the work. Let us know if further clarification is needed.

---

> > ### Comment · Reviewer_9WzV · 2024-11-25
> > **Official feedback**
> >
> > The rebuttal answered all my questions, so I kept the borderline acceptable.

---

### Official Review · Reviewer_D6gn · 2024-11-04

**Soundness:** 4
**Presentation:** 2
**Contribution:** 3
**Rating:** 6
**Confidence:** 3

**Summary:**

Authors of this work introduce ControlMM, an approach to text-to-motion generation that incorporates spatial control signals. ControlMM uses a pretrained motion tokenizer and a conditioned masked transformeras the first stage of generation. Next, it uses inference-time logit editing to refine control precision, enhancing the control precision.

**Strengths:**

* The problem of spatially controllable text-to-motion generation is an important and difficult problem.
* Results look impressive.
* Proposed methods supports a wide range of downstream applications.

**Weaknesses:**

* I found the write-up challenging to follow due to inconsistent notation in some sections and unexplained terms. The figures are also quite small with small fonts that are hard to read. They also don’t provide much additional insight. Additionally, certain details about the method are deferred to later sections, which disrupts readability. As someone less familiar with discrete latent motion representations, I needed to revisit the original MMM paper (Pinyoanuntapong et al., 2024) to understand the paper, but I still have questions about the approach. I believe the write up will benefit from further refinement in eloborating on the details of the approach and clear referencing to previous works if need be (e.g. following X, we represent Y as Z).
* In addition, one of my concerns is that authors seem to be down-playing the importance and influence of previous works which are the building blocks of the current proposed paper. After digging deeper into the paper, it looks to me as this work is using the Masked Motion Models while adding support for spatial constraints via the ControlNet mechansim. Then some additional components and tricks are applied to make the model even more effective. Despite all this, the write up seems to avoid clearly mentioning the source of the building blocks and that the works is incremental. This is one of main contributors of the lack of readability as important details are missing, but not referenced.

**Questions:**

1. There are minor typos in the text; e.g. "inference-timelogits" in line 145.
2. Figure 3 and Figure 4 are too small with very small font for the text, making it very difficult to read. If the figure doesn't fit the width of the text, maybe you should try to break it down into multiple components. I personally found the figures in the Masked Motion Models paper [Pinyoanuntapong et al., 2024] more helpful.
3. In Equation 1, what is $\mathbf{e}$? After reading the VQ-VAE paper, I understand that it's the tokenized embedding but this must be clarified for the general audience for improved readability.
4. In Equation 2, what is $L$? Also, from the description provided in the paragraph above Equation 2, it seems like $x_i$ is the $i$-th component of the quantized motion. But $x_i$ was previously used to represent the original motion frames in line 167. This choice of notation is confusing and I suggest either changing it, or emphesizing that these are different.
5. In lines 184-188, inference time iterations are discussed for the very first time. But it is very confusing as what these "iterations" really are after reading these lines. I would recommend either moving these lines to Section 3.3 entirely, or adding an overview of what the inference algorithm looks like in high level, and then adding lines 184-188 as follow-up detail.
6. To enhance understanding, I recommend adding the inference algorithm in the supplementary material.
7. The description of the motion representation (root positions or rotations) and whether and how it affects the proposed approach is also valuable. This needs to be added to the supplementary material.
8. I appreciate the quantitative and qualitative comparisons with recent diffusion-based methods of OmniControl and GMD. CondMDI (Cohan et al., 2024) is a more recent work that has improved performance over these two baselines. I am curious to where this work stands in the comparisons, and I believe it should be mentioned in the related work section.
9. In page.4, it says " We design a masked transformer
architecture to learn the conditional motion token distribution. This is the first attempt to incorporate
the ControlNet design principle (Zhang et al., 2023b) from diffusion models into generative masked
models, such as BERT-like models for image, video, language, and motion generation (Devlin et al.,
2019; Chang et al., 2022; 2023; Villegas et al., 2022)". I believe I've seen the same technique used in OmniControl, where they call the approach realism guidance. Particularly, similar to your suggested approach, they create a trainable copy of all the attention layers and connect them to the original corresponding layers using zero-initialized linear layers. Can you please alaborate on if and how the explained realism guidance is different from your suggested approach?

---

> ### Author Response · Authors · 2024-11-22
> **______Response to Reviewer D6gn______**
>
> Thank you for your time and valuable feedback. Below, we address your questions and comments:
>
> ```
> Q1. Writing
> ```
>
> **A1.** We increased the size of figures, fixed inconsistent notation in Section 3.1, added an algorithm in the supplementary material (Section A.2), provided details about root positions and rotations in Section A.3, and included CondMDI in the related work section.
>
> ```
> Q2. The importance and influence of previous works which are the building blocks of the current proposed paper
> ```
> **A2.** In addition to Section 3.2, where we discussed the ControlNet-like mechanism, we added more details in A.10 regarding the inspirations from the existing work and the difference between our work and existing solutions.  We also added the relevant references in section 3, when we explain our ControlMM solution. In addition, we add more implementation details in the supplementary materials. To clarify our contribution highlighted in the abstract,  ControlMM is the first work to enable spatial control over masked motion models so that we can simultaneously achieve high-quality motion generation with high-precision spatial control. Towards the end, we propose two key components: motion control model and inference-time logits and codebook editing.  Both schemes are fundamentally different from the existing solutions highlighted in section 2.
>
> ```
> Q3. Can you please elaborate on if and how the explained realism guidance is different from your suggested approach?
> ```
>
> **A3.** OmniControl is a controllable diffusion motion model, while ControlMM is a controllable masked motion model.   The realism guidance leveraged by OmniControl follows the ControlNet scheme originally proposed for text-to-image diffusion mode. The motion control model exploited by ControlMM is inspired by ControlNet, which has the unique features as follows.  1) **Overall working principle**: Motion Control Model achieves spatial control by modifying the underlying motion token distribution. Realism Guidance  realizes spatial control by alternating diffusion process. (2) **Architecture**: motion control model is a partial copy of the original masked motion model, where it does not possess the motion token classifier in the original model. Moreover, motion control model operates on the discrete motion tokens via effective motion tokenizer. (3) **Training paradigm**: motion control model is trained using the combination of (i) generative masking training,  (ii) consistency feedback  (iii) and training-time differential sampling. Realism Guidance is trained via the denoising process.  (4) **Inference process**:  motion control model follows the parallel token decoding and ControlNet leverages the iterative denoising.
>
> Applying a ControlNet-like approach to masked models, which operates directly in motion space is more challenging. ControlMM functions in the quantized latent space, requiring a decoder from the Motion Tokenizer. Moreover, the decoder and Masked Transformer operate in different spaces. The decoder does not support mask tokens. We address this limitation by averaging all embeddings in the codebook to approximate mask token. Moreover, we also concatenate relative motion control to solve “Ambiguity of Motion Control Signal”. We described more in Section A.9: The challenges of Motion Control Model.
>
> To directly compare Realism Guidance and our Motion Control Model performance, the table below presents results for OmniControl using only Realism Guidance without inference time guidance (Spatial Guidance) and ControlMM using only the Motion Control Model without inference time guidance (Logit and Codebook Editing). Our Motion Control Model shows better quality (FID) and more precise control.
>
>
> | Method                                   | R-Prec. Top-3 ↑ | FID ↓  | Diversity → | Foot Skat. ↓ | Traj. Err. ↓ | Loc. Err.  ↓ | Avg. Err. ↓ |
> |------------------------------------------|----------------------|--------|-------------|-----------------------|----------------------|---------------------|-------------|
> | Omnicontrol (only Realism Guidance)      | 0.691                | 0.351  | 9.506       | 0.0561                | 0.4285               | 0.2572              | 0.4137      |
> | ControlMM (only Motion Control Model)    | 0.802                | 0.128  | 9.475       | 0.0594                | 0.3914               | 0.2400              | 0.4041      |
>
> ```
> Q4. Provide additional insight.
> ```
>
> **A4.** We added more insights about how challenge and solution to integrate the novel ControlNet-like and Inference-time guidance in Masked Motion Model in supplementary
> Section A.9: The challenges of Motion Control Model
> Section A.10: Dual-Space Categorical Straight-Through Estimator
>
> ___
> Dear Reviewer ```D6gn```, If our responses address your concerns, we kindly ask you to consider reevaluating this work and improving your rating. If there are remaining issues or additional guidance you can provide, we would be more than happy to address them.

---

> ### Author Response · Authors · 2024-11-24
> **______Additional Response to Reviewer D6gn______**
>
> Dear Reviewer D6gn,
>
> Thank you for your review. As the discussion deadline is approaching in two days, could you please check our response and let us know if anything remains unclear? If your concerns are resolved, we would appreciate it if you could consider reevaluating the work. Let us know if further clarification is needed.

---

> > ### Comment · Reviewer_D6gn · 2024-11-27
> > **Response to authors' rebuttal**
> >
> > I appreciate the authors' rebuttal. I'm happy with the updated write-up now and I'm happy to keep my initial score of 6.

---

### Author Response · Authors · 2024-11-22
**______General Response to All Reviewers______**

We sincerely appreciate your time and effort in reviewing our submission and providing valuable feedback to enhance our work. We are delighted that the reviewers have acknowledged the strength of our paper such as novelty, effectiveness, superior performance, and broader applications.
In this global response, we further highlight the strengths of our paper shared by the reviewers. Since the questions brought by the reviewers have little overlap, individual questions are answered in the per-reviewer responses. We are looking forward to the valuable feedback

**Shared Strengths:**

- Novel and effective method
  - Reviewers recognizes novelty of our approach (```d9b4```, ```gWgL```) and highlight its effectiveness in controllable motion generation (```9WzV```, ```d9b4```, ```KEsH```)
- Impressive performance
  - Reviewers acknowledge that our method outperforms the SOTA method (```9WzV```, ```d9b4```) and shows impressive performance (```D6gn```)
- Well-motivated work with broad applications
  - Reviewers comment our work is well-motivated (```D6gn```) and inspiring (```d9b4```) with a wide range of applications (```D6gn```, ```9WzV```)

**Manuscript Updated**
- Added MotionLCM baseline to Figure 2. Figure 6
- Added MotionLCM CondMDI to Related work
- Added MotionLCM to https://anonymous-ai-agent.github.io/CAM/
- Updated Figure 3. and Figure 4.
- Fixed inconsistent notation in Section 3.1,
- Added an algorithm in Section A.2: Pseudo Code of ControlMM Inference
- Added details to Section A.3: Implementation Details
- Updated Section A.9: The challenges of Motion Control Model
- Updated Section A.10: Dual-Space Categorical Straight-Through Estimator
- Added KIT dataset in Section "A.12 KIT DATASET"

The update is highlight in light blue color.

---

> ### Author Response · Authors · 2024-11-26
> **______Additional Response to All Reviewers______**
>
> - Added visualizations to the "Component Analysis" section in https://anonymous-ai-agent.github.io/CAM/#component-analysis

---

### Author Response · Authors · 2024-12-04
**______Similar Questions Summary______**

**Our contribution:** ControlMM is the first to introduce controllability to Masked Motion Model through two novel components, [1] Motion Control Module (ControlNet-like) and [2] Logits/Codebook Editing (Inference Time Guidance). Moreover, ControlMM achieves SOTA in both quality and control precision, while supporting real-time generation and a wide range of applications. The visualization comparisons and applications can be found at https://anonymous-ai-agent.github.io/CAM/

> ```D6gn```, ```D9b4```, and ```KEsH```: OmniControl and MotionLCM have similar ControlNet-like components.
- While OmniControl and MotionLCM employ ControlNet-like components, both are diffusion-based models that control noise in the continuous space. The ControlNet-like approach in ControlMM utilizes Masked Motion Modelling that explicitly controls motion token distribution. This is more challenging due to its non-differentiable tokens in the latent space and corrupted masked tokens during the unmasked process. Therefore, our approach has unique network architecture, training, and inference strategies. Moreover, our ControlMM outperforms diffusion-based models in quality, precise control, while achieving real-time performance.

> ```D9b4``` and  ```KEsH```: OmniControl and MotionLCM does not use Inference Time Guidance.
- OmniControl also uses guided diffusion, which is a form of Inference Time Guidance. However, ControlMM has significantly better quality (FID) and precision control while maintaining faster generation.
- MotionLCM uses 1-step diffusion to achieve the fastest generation; therefore, implementing guided diffusion for motionLCM is not feasible. While MotionLCM is fast, its motion generation has the lowest quality and least accurate control. In contrast, our ControlMM achieves the SOTA in all metrics: quality (R-Precision, FID, and Foot Skating) and precision control (Trajectory, Location, and Average Error) while maintaining real-time generation.

_Please refer to the table in answer “A1.2” of the response to reviewer KEsH, which includes a comparison of all components of SOTA methods to provide a complete overview._

---

### Meta-Review · Area_Chair_fQE4 · 2024-12-20

**Metareview:**

The submission introduces an approach to incorporate spatial control signals into generative masked motion models.  While reviewers appreciated the results, they raised significant concerns regarding the writing and presentation, technical contributions over existing work, and evaluation. Post rebuttal, all remained unconvinced, and none would like to champion this submission.  The AC agreed with the reviewers and encouraged the authors to revise the submission accordingly for the next venue.

**Additional Comments On Reviewer Discussion:**

The rebuttal was helpful, but unable to fully convince any reviewer to champion the submission.

---

### Decision · Program_Chairs · 2025-01-22

Reject